# Additive manufacturing of an ultrastrong, deformable Al alloy with nanoscale intermetallics

Anyu Shang[1], Benjamin Stegman[1], Kenyi Choy [2], Tongjun Niu[1,3], Chao Shen[1], Zhongxia Shang[1], Xuanyu Sheng [1], Jack Lopez[1], Luke Hoppenrath[1], Bohua Peter Zhang[1], Haiyan Wang [1], Pascal Bellon [2] & Xinghang Zhang [1] ✉

Light-weight, high-strength, aluminum (Al) alloys have widespread industrial applications. However, most commercially available high-strength Al alloys, like AA 7075, are not suitable for additive manufacturing due to their high susceptibility to solidification cracking. In this work, a custom Al alloy $Al_{92}Ti_2Fe_2Co_2Ni_2$ is fabricated by selective laser melting. Heterogeneous nanoscale medium-entropy intermetallic lamella form in the as-printed Al alloy. Macroscale compression tests reveal a combination of high strength, over 700 MPa, and prominent plastic deformability. Micropillar compression tests display significant back stress in all regions, and certain regions have flow stresses exceeding 900 MPa. Post-deformation analyses reveal that, in addition to abundant dislocation activities in Al matrix, complex dislocation structures and stacking faults form in monoclinic $Al_9Co_2$ type brittle intermetallics. This study shows that proper introduction of heterogeneous microstructures and nanoscale medium entropy intermetallics offer an alternative solution to the design of ultrastrong, deformable Al alloys via additive manufacturing.

Aluminum (Al) alloys are widely utilized as structural materials in aerospace and automobile industries[1,2]. To fulfill the complex geometrical constraints for industrial applications, selective laser melting (SLM) has been increasingly used to fabricate parts of Al alloys, offering a high level of design flexibility. Most existing studies have been conducted mainly for near-eutectic Al-Si and Al-Si-Mg alloys[3]. These alloys exhibit medium strength but great hot-tearing resistance, making them good candidates for 3D printing[2,4,5]. In contrast, high-strength Al alloys, such as AA 6061[6] and AA 7075[7], are inherently susceptible to hot cracking during additive manufacturing process.

One method to alleviate hot cracking during additive manufacturing of high-strength Al alloys is to introduce fine and hard particles[5,8]. These particles can be introduced via external inoculation, e.g. TiN[9,10], TiC[11–13], TiB$_2$[14–17] or aging, e.g. Al$_3$Zr[18–20], Al$_3$Sc[21,22], Al$_2$Cu[23,24], and can strengthen the Al alloy by impeding dislocation movements.

Meanwhile, these particles promote heterogeneous nucleation, and break down columnar grains where intergranular cracks are prone to initiate and propagate[20]. In spite of these studies, the highest strength achieved in additively manufactured (AM) Al alloys remains in the range of 300–500 MPa[2]. There is scattered success in producing high strength Al alloys via severe plastic deformation[25], such as high-pressure torsion and accumulative roll-bonding, or cryo-milling followed by powder consolidation[26]. The high strength in these cases arises from significant grain refinement to nanoscales. Ultra-strong AM Al alloys with high flow strength and deformability remain to be discovered.

Transition metal (TM) intermetallics, such as Al-Fe, Al-Co and Al-Ni are largely avoided in AM Al alloys as prior experience in casting shows that the addition of TM elements often leads to large and brittle intermetallics[27,28]. These intermetallics, such as $Al_9Co_2$, $Al_{13}Fe_4$ have

[1]School of Materials Engineering, Purdue University, West Lafayette, IN 47907, USA. [2]Department of Materials Science and Engineering, University of Illinois Urbana-Champaign, Champaign, IL 61801, USA. [3]Los Alamos National Lab, Albuquerque, NM 87545, USA. ✉e-mail: xzhang98@purdue.edu

crystal structures with low symmetry (monoclinic) and thus are known to be brittle materials at room temperature. Recent studies show the addition of a small amount of Fe or Ni, can improve the mechanical strength of Al alloys, but the plastic deformability and deformation mechanisms remain unknown[28].

In this work, several transition metal elements including Co, Fe, Ni and Ti are selected to produce intermetallics-strengthened AM Al alloys. Colonies of nanoscale intermetallics lamellae aggregate into fine rosettes and give rise to a high strength, exceeding 700 MPa, with prominent plastic deformability under compression. The heterogeneous microstructure also introduces significant back stress. Complex dislocation structures and stacking faults are present in the sandwiched monoclinic brittle $Al_9(Fe,Co,Ni)_2$ phase. This study demonstrates an effective strategy to develop ultrahigh-strength AM Al alloys via nanoscale laminated deformable intermetallics.

## Results

### Microstructural characterization

Back scattered scanning electron microscopy (SEM) images reveal the microstructure of the as-printed $Al_{92}Ti_2Fe_2Co_2Ni_2$ fabricated with 300 W laser power in Fig. 1. Morphologies of inter-woven laser tracks and inverse-parabolic melt pool cross sections typical in SLM-processed alloys are evident on horizontal XY plane (Fig. 1a) and vertical XZ plane (Fig. 1b), where Z axis is the build direction. Melt pools outlined by yellow dash lines are around 120 μm in width and 80 μm in depth, with some inherent variation due to layer rotations. Figure 1c, d

show a gradient heterogeneous microstructure in the melt pools. Colonies of layered aggregates (referred to as rosettes) shown in light contrast are intermingled with the Al rich matrix (in dark contrast). Rosettes with finer laminate spacing (fine rosettes) are the dominant features near the melt pool boundaries, while rosettes with thicker lamellae (coarse rosettes) are abundant in the melt pool center. In addition, there are also some fine rosettes in the melt pool center arranged in a striated fashion outlined in pink. High-magnification SEM micrographs in Fig. 1e, f show that the melt pool center consists of coarse rosette region (36 vol.%), nanoscale cellular precipitates (4 vol.%) denoted by yellow arrows and Al rich matrix (60 vol.%). In contrast, near the melt pool boundaries, fine rosettes (97 vol.%) are separated by thin layers of Al matrix (3 vol.%).

Representative fine rosettes and cellular precipitates in coarse rosette region were characterized by scanning transmission electron microscopy (STEM) and energy dispersive x-ray spectroscopy (EDS) in Fig. 2. The fine rosettes as shown in Fig. 2a, c have $Al_3Ti$ cores surrounded by two alternating intermetallics laminates, $Al_3Ti$ and medium entropy $Al_9(Fe,Co,Ni)_2$ intermetallics, with a laminate thickness of 20–60 nm. The $Al_3Ti$ layers are thinner than $Al_9(Fe,Co,Ni)_2$ in the laminates. The chemical compositions for coarse rosettes are nearly identical to fine rosettes with a laminate thickness of 150–300 nm. The coarse rosette region also contains cellular boundaries enriched in $Al_9(Fe,Co,Ni)_2$ as shown in Fig. 2b, d.

Detailed microstructure examinations of the same (300 W) specimen using TEM and STEM are summarized in Fig. 3. The bright-field

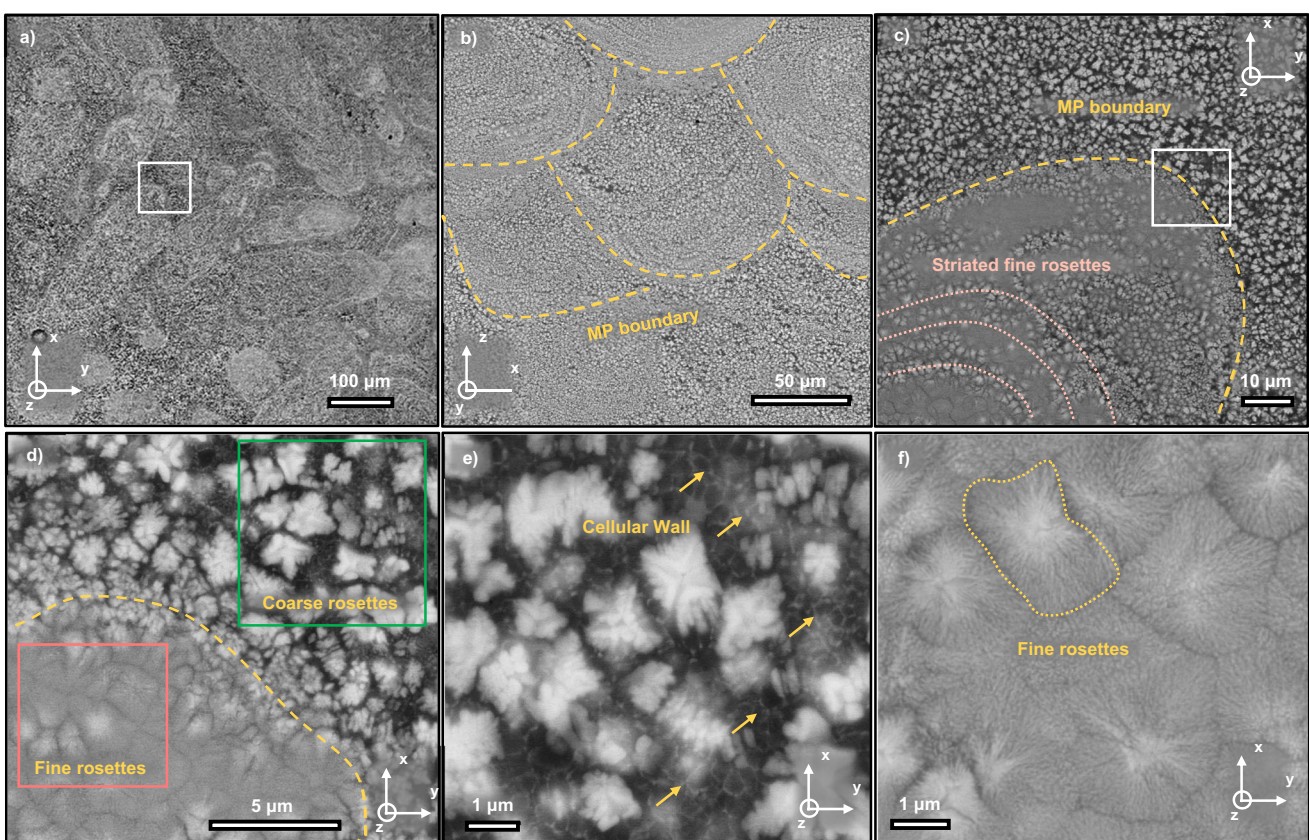

**Fig. 1 | Back scattered scanning electron microscopy (SEM) images showing overview microstructure of the as-printed $Al_{92}Ti_2Fe_2Co_2Ni_2$ alloy with 300 W laser. a** The horizontal (XY) view. The region outlined in the white box is enlarged in (**c**). **b** The vertical (XZ) planes. Z is the build direction. Yellow dash lines outline the melt-pool (MP) boundaries. **c** Micrograph of a representative melt pool with the outlined boundary and striated fine rosettes. The region outlined in the white box is magnified in (**d**). **d** The microstructure across the melt pool boundaries showing distinctive features in the coarse rosettes region and fine rosettes region. The regions outlined in the green and pink boxes are enlarged in (**e**, **f**) for coarse rosettes and fine rosettes, respectively. **e** A micrograph of the coarse rosettes region with yellow arrows indicating cellular precipitates. **f** A micrograph of the fine rosettes region showing the densely packed fine rosettes marked by yellow dotted lines with limited content of Al.

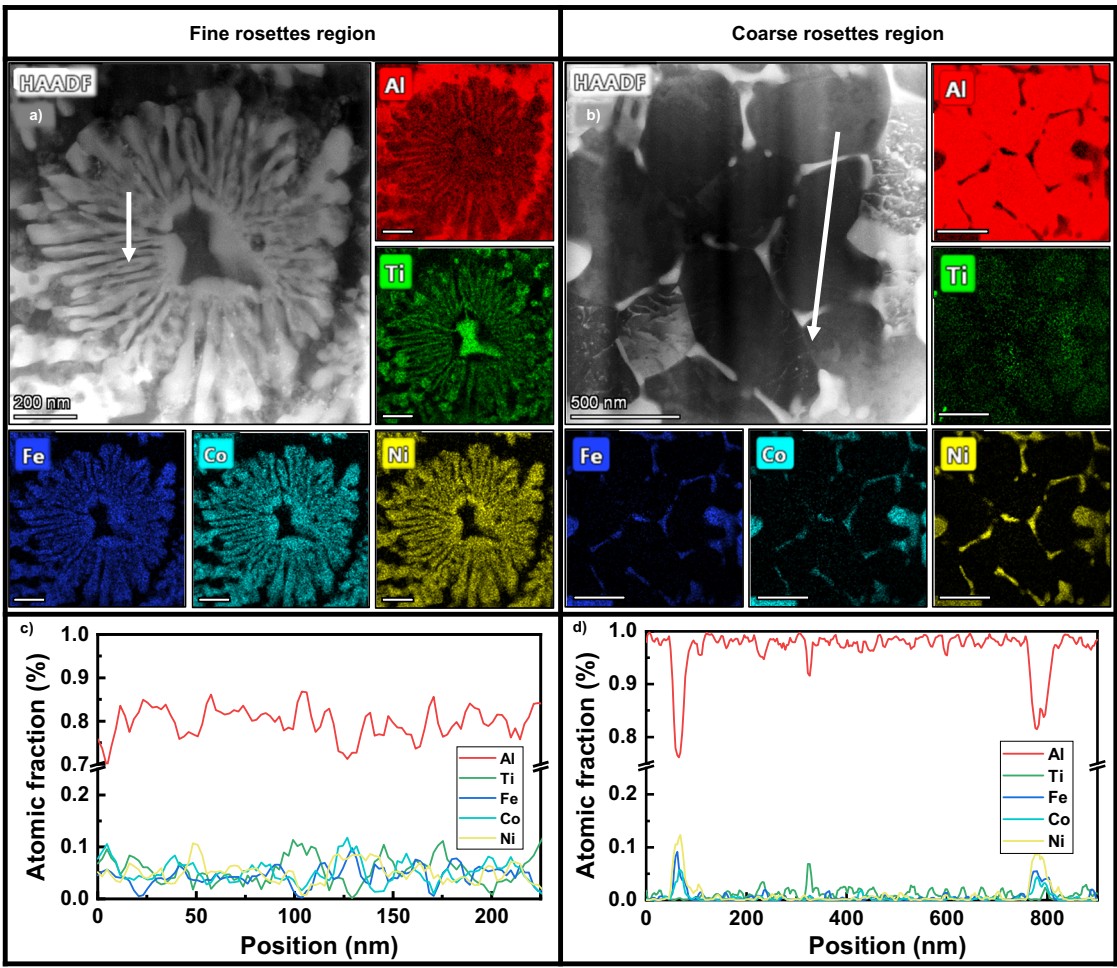

**Fig. 2 | Morphology and compositions of nanoscale intermetallics.** High-angle annular dark-field scanning transmission electron microscopy (HAADF-STEM) images showing representative morphologies of (**a**) fine rosettes region and (**b**) coarse rosettes region and corresponding energy-dispersive x-ray spectroscopy (EDS) composition maps of various elements. The line scans in (**c**, **d**) showing relevant phase constituents following the white arrows in (**a**, **b**), respectively. The fine rosettes have an $Al_3Ti$ core with alternate lamellae composed of $Al_3Ti$ and $Al_9(Fe,Co,Ni)_2$. The cellular precipitates in coarse rosettes region are composed of $Al_9(Fe,Co,Ni)_2$.

(BF) TEM image in Fig. 3a shows the coexistence of three major phases, Al matrix (red arrows), $Al_9(Fe,Co,Ni)_2$ (orange arrows), and $Al_3Ti$ (green arrows). X-Ray diffraction (XRD) pattern (Supplementary Fig. 1) and selected area electron diffraction (SAED) of the TEM image (Supplementary Fig. 3) suggest the $Al_3Ti$ exists mainly in $DO_{22}$ phase (space group I4/mmm, a = 0.384 nm, c = 0.860 nm[29]) and partially in $L1_2$ phase (space group Pm3̄m, a = 0.398 nm[30]), and $Al_9(Fe,Co,Ni)_2$ has monoclinic structure with the prototype of $Al_9Co_2$ (space group $P2_1/c$, a = 0.622 nm, b = 0.629 nm, c = 0.856 nm, β = 94.8° [31]) or $Al_9FeNi$[32]. Besides, it could be seen that plenty of defects exist in $Al_3Ti$ (Fig. 3a & Supplementary Fig. 3). The inverse pole figure (IPF) map in Fig. 3b acquired from high-resolution ASTAR orientation mapping by high-precision electron diffraction recognition[33,34] demonstrates the crystallographic orientation of Al and $Al_3Ti$. The polycrystalline composite has colonies with dimension of ~ 1 μm. High-resolution TEM (HRTEM) image (Fig. 3c) along $Al_3Ti$ $DO_{22}$ [100] zone axis demonstrates the lattice distortion induced by the 2 nm scale scattered patches. An inverse Fast Fourier Transform (FFT) image based on (002) diffraction shows plenty of dislocations residing in the disordered lattice patches as shown in Fig. 3d. In comparison, the $Al_9(Fe,Co,Ni)_2$ phase possesses few defects as shown in Fig. 3a. High-resolution STEM (HRSTEM) (Fig. 3e) shows the atomic arrangements of the medium-entropy $Al_9(Fe,Co,Ni)_2$ along [110] zone axis with blue dots representing sites of Co atoms in its prototype. The micrograph is consistent with the

simulated $3 \times 3 \times 3$ cells by VESTA[35] (Fig. 3f). It's difficult to distinguish Fe, Co, Ni atoms in the current HRSTEM micrograph due to their close proximity in atomic number. Figure 3g depicts the typical well-defined interface between $Al_9(Fe,Co,Ni)_2$ and $Al_3Ti$. The SAED pattern (Fig. 3h) illustrates the crystallographic orientation relationships between these two phases, where $[1\bar{3}0]_{Al_3Ti}$ // $[100]_{Al_9(Fe,Co,Ni)_2}$, $(002)_{Al_3Ti}$ // $(001)_{Al_9(Fe,Co,Ni)_2}$, $(310)_{Al_3Ti}$ // $(010)_{Al_9(Fe,Co,Ni)_2}$ and their interplanar spacing has the following match ratio: $d^{(002)}_{Al_3Ti} : d^{(001)}_{Al_9(Fe,Co,Ni)_2} \approx 1{:}2$ and $d^{(310)}_{Al_3Ti} : d^{(010)}_{Al_9(Fe,Co,Ni)_2} \approx 1{:}5$. The lattice mismatch values estimated from diffraction patterns for these matching planes are 2.5% and 2.1%, respectively.

Figure 4a corresponds to a TEM image of three Al matrix grains from a fine rosette region to gauge the microstructure before atom probe acquisition. Atomic maps for individual elements (Fig. 4b) show that there are some inhomogeneities in the distribution of the TM solute elements. This variation is highlighted in the 1D composition distribution in Fig. 4c, where the profile along the cylinder in Fig. 4b. Ti concentration increases from 0.1 to 0.5 at% towards the top grain with a decrease in Ni concentration concomitantly. Grain boundaries appear to enrich in Fe, Co and Ni. In general, the transition metal elements have very low solid solubility in Al: Ti < 0.05 at%, Fe: < 0.04 at %, Co < 0.05 at%, Ni < 0.04 at%. Hence the APT composition analysis in

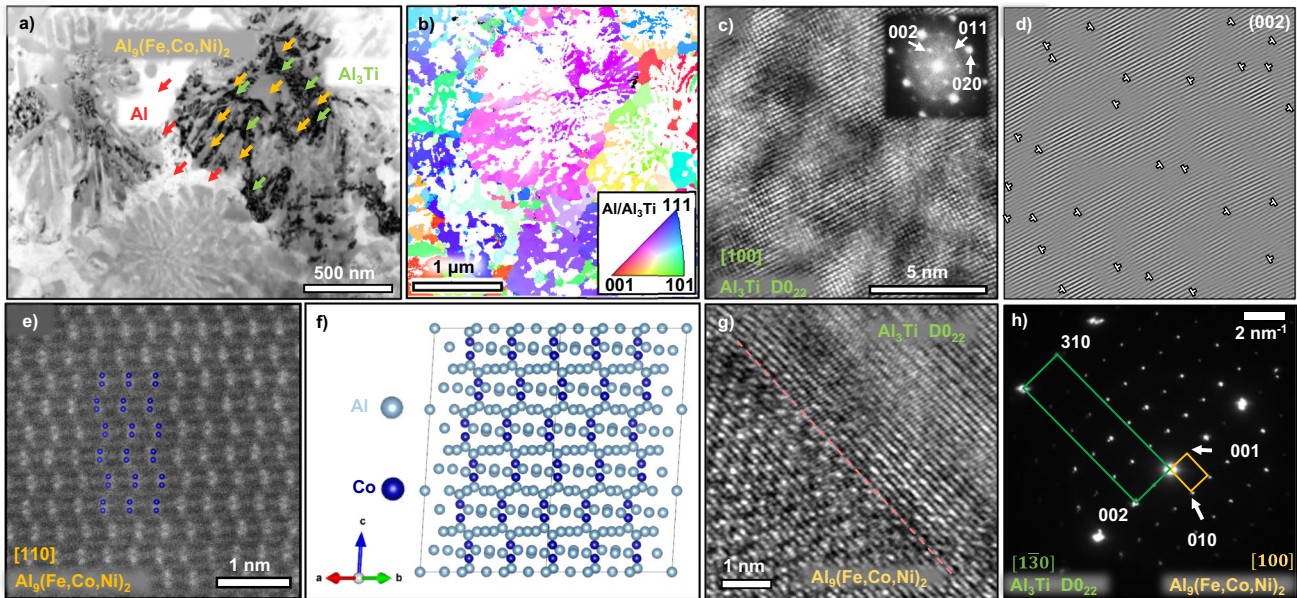

**Fig. 3 | TEM characterizations of the as-printed Al$_{92}$Ti$_2$Fe$_2$Co$_2$Ni$_2$. a** An overview transmission electron microscopy bright field (TEM-BF) image showing three major phases in the alloy, Al matrix (red arrows), Al$_9$(Fe,Co,Ni)$_2$ (orange arrows), and Al$_3$Ti (green arrows). **b** Inverse pole figure mapping of Al/Al$_3$Ti phases by ASTAR (nano-EBSD based on high-precision electron diffraction patterns). **c** High-resolution TEM (HRTEM) with the corresponding Fast Fourier Transform (FFT) of the image of Al$_3$Ti intermetallic showing its highly defective nature. **d** An inverse FFT of the image with (002) plane filtered indicating the presence of abundant dislocations in the nanoscale Al$_3$Ti intermetallics. **e** High-resolution STEM (HRSTEM) showing the atomic arrangement of Al$_9$(Fe,Co,Ni)$_2$ intermetallic compound superimposed with cobalt atoms. **f** Identical structures obtained from VESTA[35] crystallographic visualization of its prototype Al$_9$Co$_2$ with monolithic crystal structure. **g** HRTEM micrograph of the interface between two genres of intermetallics, and (**h**) the corresponding selected area electron diffraction (SAED) pattern indicating the orientation relationships.

Fig. 4c suggests that Ni and Ti concentrations in Al (0.1–0.6 at.%), have largely exceeded their equilibrium solid solubility, presumably due to supersaturation from the rapid solidification process. Figure 4d corresponds to a dual-phase interface between Al and Al$_9$(Fe,Co,Ni)$_2$ taken from another area. A relatively small amount of TM solutes are present in the matrix, specifically, 0.20% Ti, 0.30% Fe, 0.03% Co and 0.09% Ni (at%). Ti is highly rejected by the intermetallic phase and present in the matrix. The 1D composition distribution profile in Fig. 4e along the cylinder shown in Fig. 4d shows minute TM solutes in matrix, vs. nearly equiatomic distribution of Fe, Co and Ni (78.06% Al, 0.06% Ti, 5.45% Fe, 7.53% Co and 8.90% Ni (at%)) in the intermetallic phase. To understand the interactions between the alloying elements in the brittle intermetallic region of the tip, partial radial distribution functions (Fig. 4f) were computed (in the area highlighted by the yellow rectangle). All profiles remain close to 1.0, the value for random distribution, signifying well-mixed TM solutes in lattice. Another tip on the Al$_9$(Fe,Co,Ni)$_2$/Al$_3$Ti interface was examined by APT to confirm the chemical constituents of both intermetallic phases (Supplementary Fig. 4). There is no evidence of the presence of Al at the interfaces, suggesting rosettes are mostly intermetallics.

## Mechanical properties
In order to assess the mechanical properties of heterogenous AM Al alloys, nanoindentation experiments were conducted over a representative melt pool (Fig. 5a). Hardness contour reconstructed from nanoindentation map (Fig. 5b) shows a majority of the region has a high hardness ranging from 2.5 to 4.5 GPa. Melt pool boundaries often have a higher hardness (3.5–4.5 GPa), whereas the melt pool interior has a lower hardness (2.5–3.0 GPa). A similar trend for Young's modulus is observed. Hard melt pool boundaries are associated with a relatively high Young's modulus (140–150 GPa), while melt pool interiors have a relatively low Young's modulus (130–140 GPa).

Bulk compression tests were performed on cylinders with dimensions of 6 × 12 mm, fabricated at various laser power. Specimens printed with 200 W laser (red) exhibit an ultra-high engineering stress exceeding 800 MPa concurring with substantial plastic deformability around 20%. The inserted optical micrographs reflect the typical barreling phenomenon for ductile metallic materials. At higher laser energy, 250 W and 300 W, the flow stress of pillar decreases to 550 MPa with compressive strain of 5–20%. Figure 6b presents analyses on plastic instability by using Considère's criterion. The superimpositions of selective work hardening curves over true stress – true strain curves imply uniform deformation strain at 7%.

To probe the influence of heterogeneous microstructures on mechanical behavior of the AM Al alloys, micropillar compression tests were carried out over coarse and fine rosettes regions. As shown in Fig. 7a, fine rosette region can sometimes possess a high strength exceeding 900 MPa, while coarse rosette region has a flow stress of 500 MPa (Supplementary movies 1 and 2). In situ SEM micrographs in Fig. 7b collected from supplementary movies reveal the morphological evolution for representative pillars from these two distinct regions. Rosette precipitates readily visible on the pillar surfaces in the coarse rosette region due to axial compression, while a wrinkled and wavy surface shows less apparent protrusion in the fine rosette region. For the coarse rosettes, the evident rosette precipitate extrusions to the outer surfaces manifest the preferential plastic flow in the soft aluminum matrix and insignificant plasticity in the hard intermetallics precipitates. Whereas the retention of cylindrical shape of pillars in the fine rosettes implies uniform co-deformation of matrix and precipitates to constrain the generation of localized shear bands. The multiple loading-unloading experiments display prominent hysteresis loops evident from the stress-strain curves. The implication of such loops on deformation mechanisms will be discussed in the discussion section.

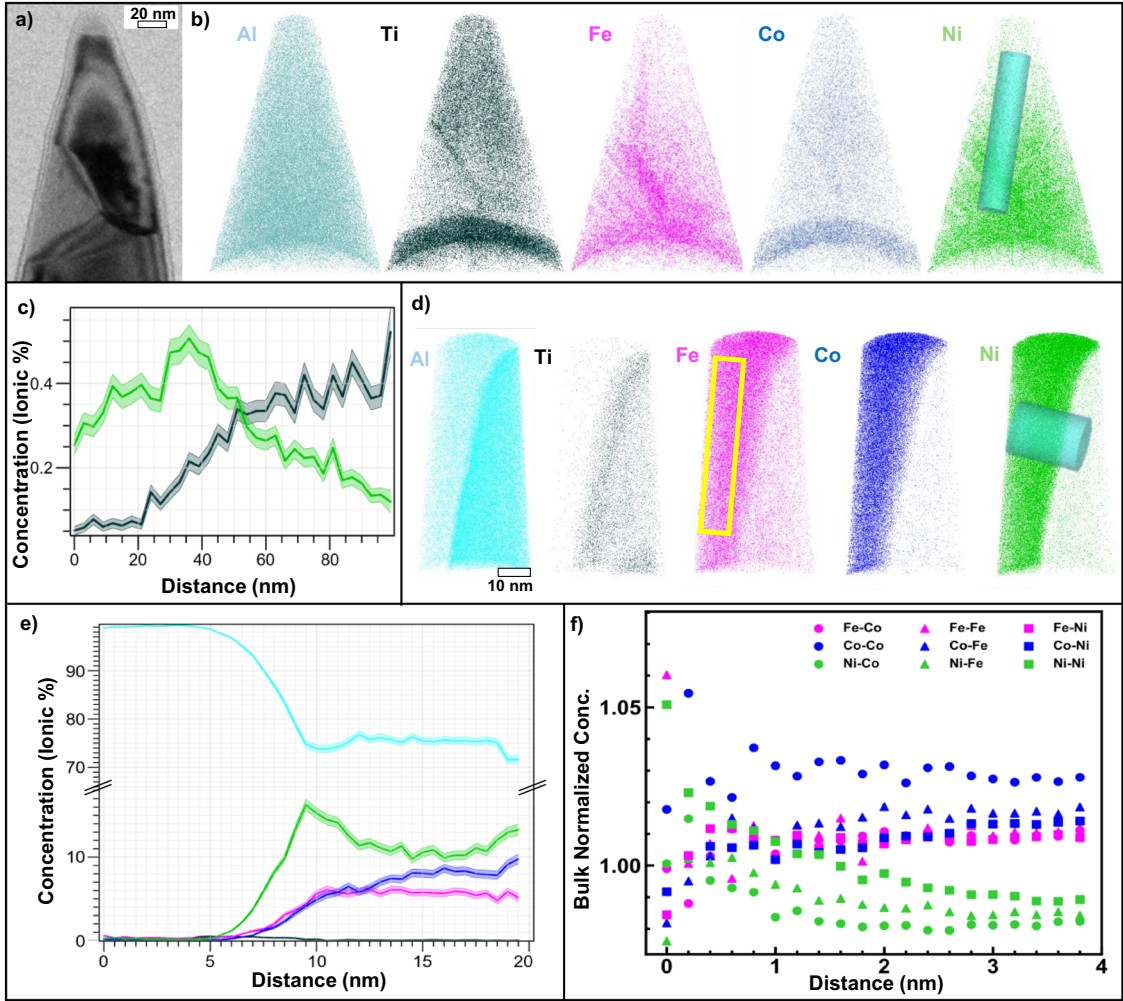

**Fig. 4 | Atom Probe Tomography (APT) analyses. a** An image of atom probe tip taken prior to APT acquisition, demonstrating a three-grain region. **b** Atomic maps demonstrating uneven saturation of solute elements. **c** 1D concentration profile (obtained from bottom to top of the cylindrical region of interest on Ni map in (**b**) capturing the behavior of the solute elements across two grains. The concentrations of Ti, Fe, Co, and Ni in these three Al grains are below 1 at.%. Notice also in (**c**) some segregation at grain boundaries, as well as the variability of solute concentration inside the different grains, e.g., Ni and Ti. Error bars display standard error for plotted concentrations from the sampled region of interest. **d** Atomic maps of a sample demonstrating an Al grain saturated with all solute elements, and the monoclinic $Al_9(Fe,Co,Ni)_2$ phase. **e** 1D concentration profile taken across the interphase boundary following the cylindrical region on Ni map in (**d**). The profile highlights a transition from a dilute Al grain on the right side, to the $Al_9(Fe,Co,Ni)_2$ phase on the left. Error bars display standard error for plotted concentrations from the sampled region of interest. **f** Radial Distribution Function (RDF) taken from the yellow inset in (**d**) highlights the absence of solute clustering in the bulk monoclinic phase.

## Post-deformation microstructure analyses

To better understand the deformation mechanism of AM Al alloys with intermetallic rosettes, cross-section TEM (XTEM) samples from post-deformation micropillars were investigated. In the coarse rosette region (Fig. 8a), some microcracks are witnessed in intermetallics in STEM image. EDS elemental maps are provided in Fig. 8b–d to identify composition in these intermetallics precipitates. Microcracks (labeled in dash circles) appear in both $Al_3Ti$ and $Al_9(Fe,Co,Ni)_2$ intermetallics. It's witnessed that these microcracks are typically constrained in single intermetallic phase, not extending into the neighboring phases. Figure 8e, f show abundant dislocation activities in the vicinity of cellular walls in Al matrix.

In the fine rosette region, the dilation of the pillar top and shear bands were observed (Fig. 9a). Apart from dislocation activities and microcracks, significant crystallographic rotation (as evidenced by the inserted SAED pattern) and some planar defects were observed in the monoclinic $Al_9(Fe,Co,Ni)_2$ phase (Fig. 9b). Figure 9c shows the $Al_9(Fe,Co,Ni)_2$ precipitate (outlined in red) has an evident curvature with planar defects and grain rotation on the pillar top, where the deformation is the most intense. The HRTEM micrograph of the

$Al_9(Fe,Co,Ni)_2$ shows abundant stacking faults (Fig. 9d), and streaks in the inserted FFT pattern suggest the habit plane for these SFs is (001). The related inverse FFT image masking two brightest diffraction spots reveals dislocations aligned along the SFs (Fig. 9e). Another HRTEM micrograph (Fig. 9f) was acquired for $Al_9(Fe,Co,Ni)_2$ at 400 nm underneath the top surface, where the deformation is less severe. The lattice arrangement is disturbed by (001) SFs along [110] zone. Streaks in the indexed FFT pattern are pronounced and extra spots circled in pink are identified. Figure 9g depicts the ends of a SF ribbon. The corresponding inverse FFT image in Fig. 9h confirms the additional spots originate from the faulted region, which might suggest a possible phase transformation, as the interplanar spacing has no matching with the parent phase. Figure 9i demonstrates a low-angle grain boundary (~5°) within $Al_9(Fe,Co,Ni)_2$. The boundary is roughly on $(1\bar{1}0)$ where some local disorder is present.

## Discussion

A highly heterogeneous microstructure composed of coarse rosettes, fine rosettes and cellular Al matrix was observed in the AM Al alloy. To further understand the formation of rosettes, equilibrium phase

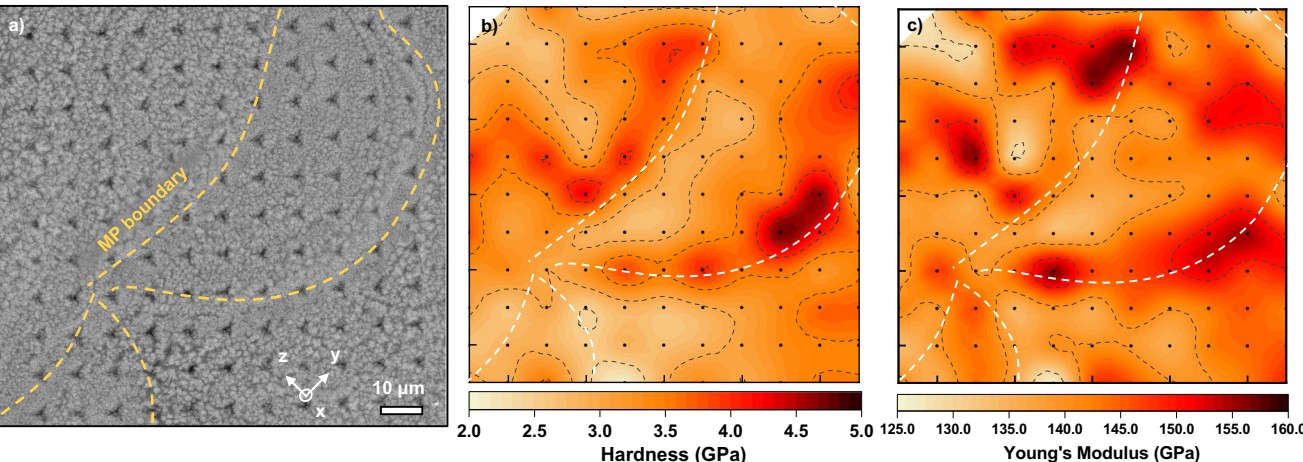

**Fig. 5 | Hardness contours derived from nanoindentation results. a** A back scattered scanning electron microscopy (SEM) micrograph on the as-printed $Al_{92}Ti_2Fe_2Co_2Ni_2$ after nanoindentation hardness measurements. **b** The hardness and (**c**) Young's modulus contour maps re-constructed from series of nanoindentation, revealing a relatively higher hardness/ Young's modulus near the melt pool boundaries with finer microstructure.

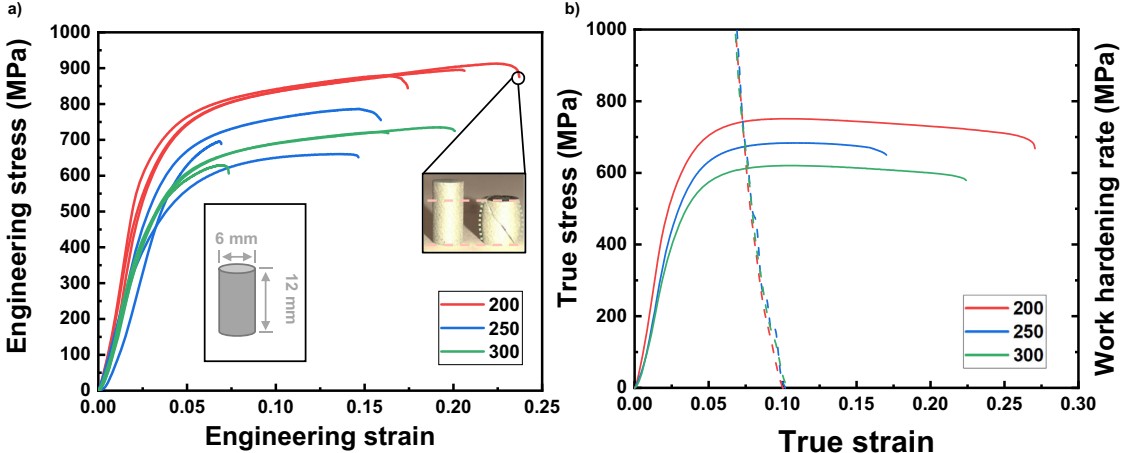

**Fig. 6 | Macroscale mechanical properties under compression tests.**
**a** Engineering stress – engineering strain curves for bulk compression tests performed on the as-printed $Al_{92}Ti_2Fe_2Co_2Ni_2$ pillars with inserted schematic diagrams showing the geometry of specimen and the typical barreling phenomenon after compression for one of the deformed specimens (200 W). Numbers in the legend denote laser power in Watts with three repetitions in each condition for reproducibility. **b** True stress - true strain curves (solid lines) superimposed with work hardening rates (dashed lines) showing 7% uniform compression strain in the early stages of deformation followed by long toughing stages.

constitution was calculated by Thermo-calc using TCAL8 database (Supplementary Fig. 5). The calculation suggests that $Al_3Ti$ forms the cores of intermetallic rosettes due to its high melting temperature, providing nucleation sites for co-precipitation of $Al_9(Fe,Co,Ni)_2$. The rapid cooling rate significantly refines precipitates, whereas traditional casting of transition-metal-bearing Al alloys often leads to overgrown large precipitates and hence, embrittlement. The morphology distinctions for coarse and fine rosettes are attributed to the complex and location-specific thermal history with respect to melt pools. It is postulated sufficient supplies of TM solutes and a higher quenching rate adjacent to melt pool boundaries enable the precipitation of a greater volume fraction of intermetallics with finer lamellae, compared to the coarse rosettes that dominate melt pool center. The arrangement of alternating fine and coarse rosettes region in this alloy results from periodic thermal cycles during layer-wise construction. Additional refinement is realized by the striated precipitation possibly due to the turbulent Marangoni flow[36]. Marangoni flow stirs fine rosettes owing to the complex thermal gradient, varying surface tension and dynamic hydromechanics[36]. Besides, the lattice matching between $Al_3Ti$ and $Al_9(Fe,Co,Ni)_2$ may reduce the interfacial energy, and promote the formation of nanolaminated intermetallics. Under the assumption of quasi-equilibrium condition, the crystal will exhibit anisotropic Wulff shapes to reduce interfacial energy, promoting the formation of refined intermetallic rosettes with larger interfacial area. The formation of high-melting-temperature intermetallics in the melt pools may promote heterogeneous nucleation of Al grains during subsequent solidification and thus reduce the grain size of Al. Similar effects have been reported for Al alloys with the aid of nucleants, such as $Al_3Zr$[18–20], $Al_3Sc$[21,22], $Al_3Ti$[37], $TiB_2$[14–17], for the refinement of grain size of Al matrix. The rosette structure was reported in other Al alloys where Ce and Mn were introduced for precipitate strengthening, yet the resultant deformation mechanisms remain unexplored[38–41].

The high quenching rate characteristic of laser fusion not only refines the microstructure but also has profound impact on the formation of various non-equilibrium phases. First, $L1_2$ $Al_3Ti$ is often unstable and will spontaneously transform to equilibrium $DO_{22}$ $Al_3Ti$[42].

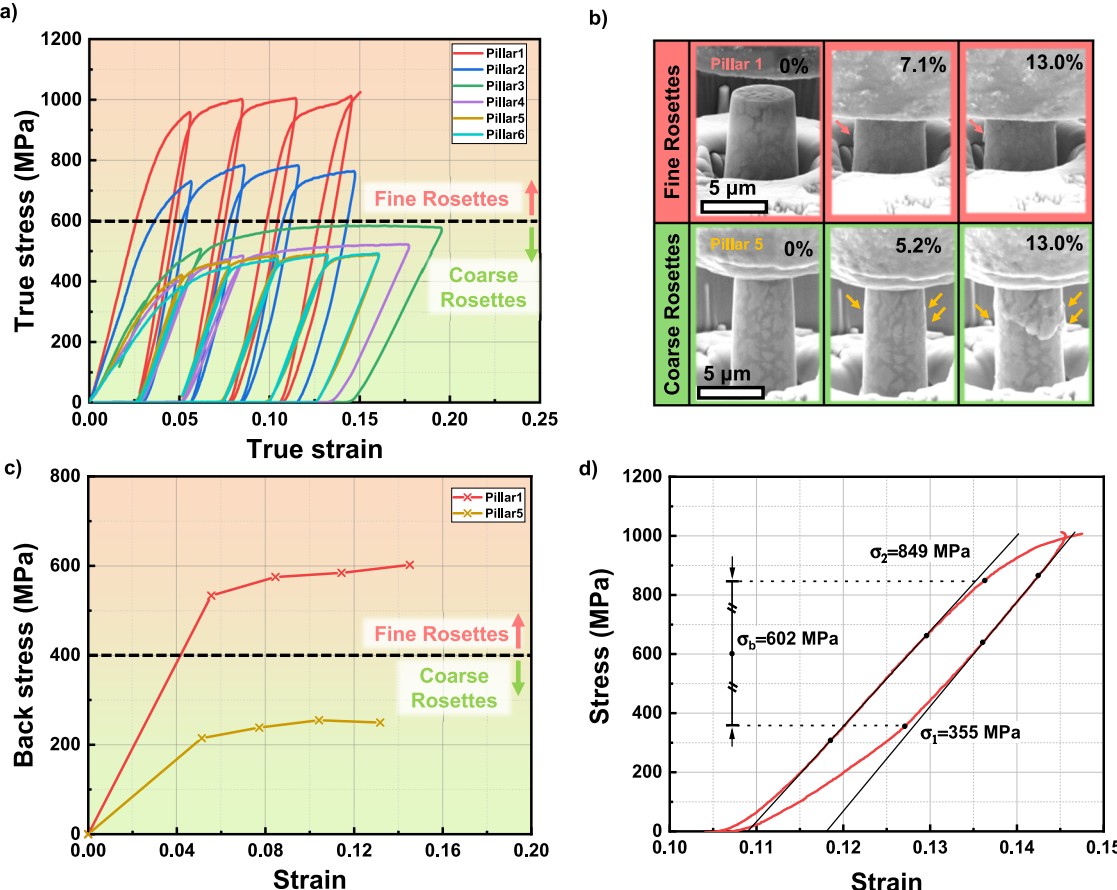

**Fig. 7 | Microscale mechanical properties revealed by micropillar compression tests. a** True stress – true strain curves for in situ micropillar compression tests on both fine (pink) and coarse (green) rosettes regions of samples printed with 300 W laser power tested at room temperature. The flow stress on fine rosette region could reach 1 GPa. **b** Screenshots of morphological evolutions for pillars under compression. Arrows indicate the formation of shear planes. **c** Back stress measurements vs. strain curves for micropillars with coarse and fine rosettes showing the prominent back stress. **d** An exemplar back stress determination based on a stress – strain hysteresis loop for the pillar in fine rosette region at 14% strain. Solid black curves based on elastic loading/unloading stages are derived from the linear sections and extrapolated to the yield points with 0.1% proof strain. The back stress value, $\sigma_b$, the yield stress for unloading, $\sigma_1$ and the yield stress for loading, $\sigma_2$ are labeled for illustration.

But the rapid solidification process retains some L1$_2$ Al$_3$Ti (Supplementary Fig. 3) by not allowing atoms sufficient time for complete ordering. The cruciform geometry of Al$_3$Ti core in Fig. 2 correlates well with literature reports on typical morphology characteristics of trialuminides[43,44]. L1$_2$ Al$_3$Ti can also be fabricated via mechanical alloying with or without a ternary element[30,45]. It is generally accepted that L1$_2$ Al$_3$Ti shall be more deformable than its D0$_{22}$ counterpart as the former has more independent slip systems rendered by a cubic crystal structure. It is also speculated that the high cooling rate establishes significant defects in both phases. Second, a partitioned medium-entropy intermetallic phase Al$_9$(Fe,Co,Ni)$_2$ was maintained, while at equilibrium (Supplementary Fig. 5) it shall decompose into two isomorphic monoclinic phases (Al$_9$Co$_2$ + Al$_9$FeNi). The supersaturated Fe, Ni atoms in Al$_9$(Fe,Co,Ni)$_2$ distort the lattice locally and thus change its mechanical behavior. Preservation of these metastable phases would play a significant role in deformation mechanisms of the AM Al alloys.

High strength of the current AM Al alloy is confirmed by multiple experiments. This alloy exhibits over 800 MPa engineering stress from macroscale compression tests. Micropillar compression tests show that the fine rosette region could reach 1 GPa true flow stress or an engineering stress of 1.18 GPa. An estimation based on the rule of mixture is attempted as shown in Supplementary Table S1[29,46,47]. Hardness assessments from nanoindentation mapping show values of 2.5–4.5 GPa. The variations of hardness values across melt pools arise from the heterogenous microstructures consisting of coarse rosettes

in melt pool center and fine rosettes near melt pool boundaries. The current Al$_{92}$Ti$_2$Fe$_2$Co$_2$Ni$_2$ has an excellent combination of mechanical strength and plastic strain under compression, compared with other AM Al alloys shown in Supplementary Fig. 6[24,48–51].

Next, we consider the related strengthening mechanisms leading to the ultrahigh mechanical strength in our AM Al alloys, including solid solution strengthening and Orowan strengthening, Hall-Patch strengthening, dislocation strengthening, and hetero-deformation induced (HDI) strengthening[52]. Solid solution strengthening can be ignored as the accumulative solubility of TM solutes in Al (though in supersaturated state) is very low, <1 at% based on EDS measurements (Fig. 2) and APT studies (Fig. 4c). Dislocations in the as-printed state do not play a significant role in strengthening the alloy. TEM experiments (Supplementary Fig. 7) show a moderate dislocation density $\rho_{disl}$ $4.7 \times 10^{13}$ m$^{-2}$ ~ $1.0 \times 10^{14}$ m$^{-2}$. Therefore, strengthening contribution from these dislocations can be estimated to be 25–36 MPa by:

$$\sigma_{disl} = \beta MGb\sqrt{\rho_{disl}} \qquad (1)$$

where β is a material constant, M is the Taylor factor, G is shear modulus, b is the burgers vector of Al. Their values are given as follows, β = 0.16, M = 3.06, G = 26.5 GPa and b = 0.286 nm[23,53]. Orowan strengthening is also secondary as the morphology of dislocations are not those typical to Orowan strengthening, which would be manifested by dislocation loops encircling intra-granular precipitates. In the

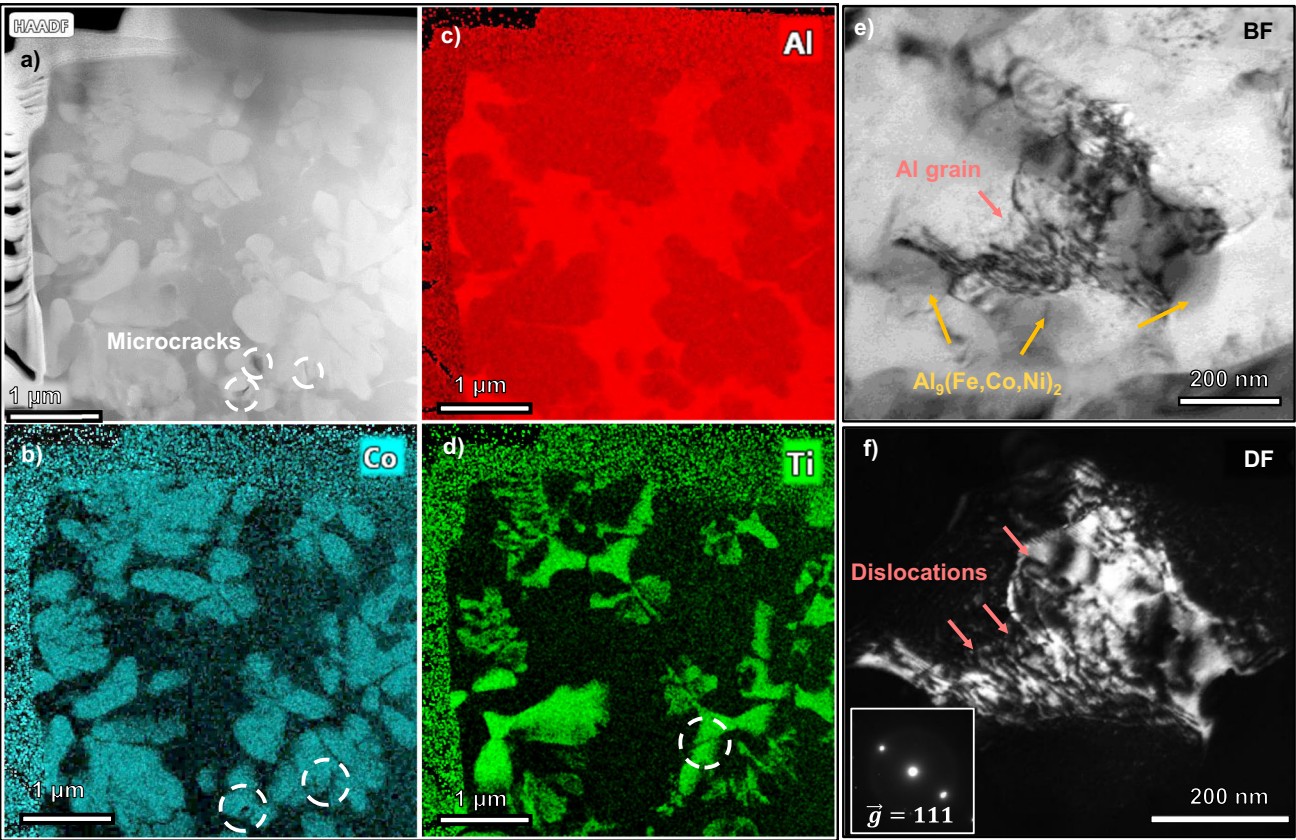

**Fig. 8 | Deformation mechanisms in coarse rosette regions. a–d** Scanning transmission electron microscopy (STEM) images and selective energy dispersive x-ray spectroscopy (EDS) mapping on the post-deformation pillar in the coarse rosette region. **e** BF and (**f**) weak-beam dark field TEM micrographs of the Al grain revealing high-density dislocations (denoted by red arrows) in the vicinity of cellular precipitates. The insert in (**f**) shows the diffraction spots after construction of weak beam condition g(3 g), g = 111.

current AM Al alloy, dislocation entanglements are primarily identified against inter-granular Al$_9$(Fe,Co,Ni)$_2$ cellular precipitates in the post-deformation TEM samples (Fig. 8e, f).

HDI strengthening has been observed in heterogenous materials and can provide back stress and work hardening ability in metallic materials[52,54,55]. Significant stress-strain hysteresis loops were observed during micropillar compression tests (Fig. 7a). The evolution of accumulated back stress with progressing strain in Fig. 7c reveals that the fine rosette regions with a higher flow strength carry a very high back stress, ~600 MPa, compared to the back stress of ~250 MPa in the coarse rosette regions. Back stress values, $\sigma_b$, are determined following the classic method[55] and are demonstrated in Fig. 7d with the following equation:

$$\sigma_b = \frac{\sigma_1 + \sigma_2}{2} \qquad (2)$$

where $\sigma_1$ and $\sigma_2$ are the yield stress values on the unloading stage and loading stage, respectively. Back stress is the long-range stress component typically ascribed to the pileups of geometrically necessary dislocations (GNDs) in materials with heterogeneity or gradient structures. These GNDs could arise from mismatch of coefficients of thermal expansion (CTE) between Al and intermetallics during solidification[56,57], and strain incompatibility across interfaces of hard and soft phases during compression. It's worth mentioning that HDI strengthening is the major component leading to Bauschinger effect as GND pileups near interfaces have reversible dislocation configuration during loading-unloading processes. TEM study reveals

ample GNDs in Al matrix shown in Fig. 8e, f. Rigid intermetallics tend to remain elastically deformed while Al matrix sustains high-density dislocations near the interfaces to accommodate strain field gradient. Existing GNDs will hamper further dislocation motion and also trigger forward stress in the hard intermetallic phases across interfaces. Back stress and forward stress will collectively harden the material. As the deformation progresses, back stress rises quickly and reaches a plateau for both regions. This trend can be explained by the absorption of GNDs by the interfaces after straining to a critical value[58]. At certain strain levels, dynamic generation and annihilation of GNDs reach an equilibrium state, leading to a saturated strengthening contribution from HDI stress. The strain gradient carried by absorbed GNDs, though decoupled from strengthening, will trigger Al-intermetallics interface debonding in microscale and eventually fracture at macroscale[58].

Apart from HDI stress from Al-intermetallic interface, there might be HDI stress originating from the interfaces between two genres of intermetallics Al$_9$(Fe,Co,Ni)$_2$ and DO$_{22}$-Al$_3$Ti. Prior study suggests both intermetallics phases are brittle at room temperature[29,59]. This assertion is especially applicable to Al$_9$(Fe,Co,Ni)$_2$ inferred from its monoclinic crystal structure with low symmetry. However, abundant SFs and dislocations were observed in the deformed nanoscale monolithic Al$_9$(Fe,Co,Ni)$_2$ (Fig. 9), suggesting unique plastic deformation mechanism in the often brittle intermetallics. This study presents what may be the first experimental evidence of plasticity in monoclinic Al$_9$(Fe,Co,Ni)$_2$ medium entropy intermetallics. There are limited studies showing the formation of complex intermetallics in AM Al alloys[39–41,60,61]. Some prior studies also suggest that complex metallic compound could deform by introducing metadislocations[62,63], which

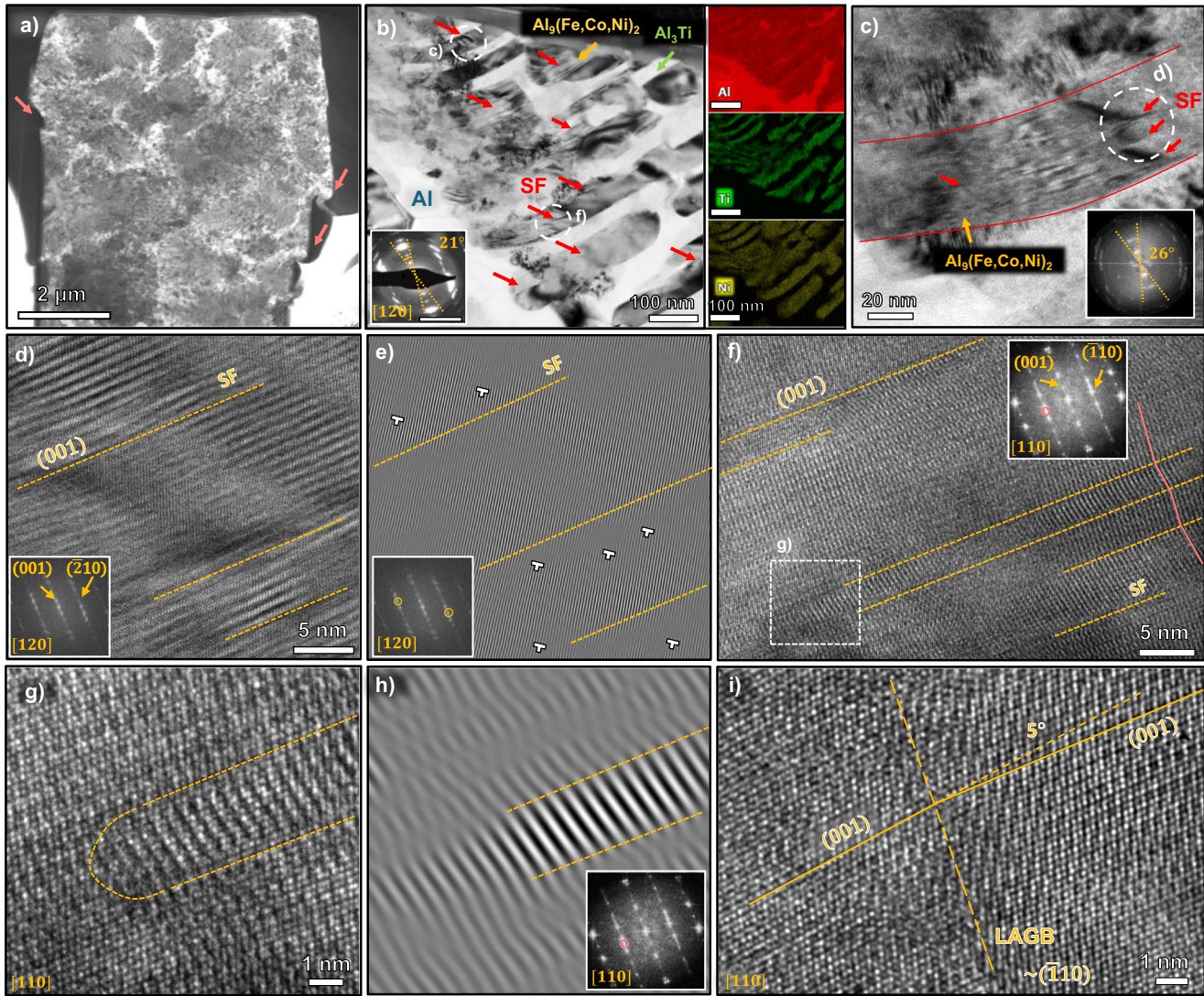

**Fig. 9 | Deformation mechanisms in fine rosette regions. a** An overview transmission electron microscopy (TEM) micrograph on the post-deformation pillar in the fine rosette region. Pink arrows indicate the deformation bands. **b** A TEM micrograph demonstrating the morphology of fine intermetallic rosettes. Phases are labeled (blue for Al, green arrows for $Al_3Ti$ and yellow arrows for $Al_9(Fe,Co,Ni)_2$) and energy dispersive x-ray spectroscopy (EDS) maps are presented. Red arrows indicate the abundant stacking faults (SFs) in the monoclinic $Al_9(Fe,Co,Ni)_2$. The insert shows the selected area electron diffraction (SAED) pattern along $Al_9(Fe,Co,Ni)_2$ [120] zone with the significant crystal rotation by plastic strain. The scale bar corresponds to 5 nm⁻¹. **c** A TEM image for $Al_9(Fe,Co,Ni)_2$ outlined in red near the pillar top in (**b**) with the corresponding Fast Fourier Transform (FFT) pattern. Stacking faults and crystal rotation are observed. **d** An High-resolution

TEM (HRTEM) micrograph for the $Al_9(Fe,Co,Ni)_2$ particle in (**c**) with planar defects. The corresponding FFT pattern reveals stacking faults habit plane is (001). **e** An inverse FFT image masking circled diffraction spots shows lattice distortion around stacking faults with some additional half planes, indicating the existence of dislocations. **f** An HRTEM micrograph for the $Al_9(Fe,Co,Ni)_2$ particle in (**b**) slightly off the pillar top with planar defects. The indexed FFT pattern along [110] zone shows diffraction spots originating from defects circled in pink. Pink solid line segments help to visualize the direction change of crystal planes. **g** An HRTEM micrograph on the end of a stacking fault ribbon. **h** An inverse FFT image masking the extra spots in the inserted FFT confirms they result from the stacking fault ribbon. **i** An HRTEM image for $Al_9(Fe,Co,Ni)_2$ showing a low-angle grain boundary (LAGB) - 5° close to (1̄10) plane where atomic disorder is observed.

rarely exist in high symmetry metallic materials, and metadislocations have been reported in $Al_{13}Co_4$ with monoclinic crystal structure[64]. There are dislocations in $Al_3Ti$ in the as-printed state (Fig. 3a−c). Hence, we speculate that dislocations should also carry out plastic flow in $Al_3Ti$ to ensure co-deformation between the two types of intermetallics across the laminated intermetallics interfaces in the fine rosette region (Figs. 2a and 9). Deformability and deformation mechanisms for nanoscale sandwiched intermetallic branches could differ from their bulk counterpart, due to the discrepancy in scale and confined loading state, as corroborated by studies on laminated nanolayers[65–68]. Comparing to transient and reversible GND pileups adjacent to Al-intermetallic interfaces, temporary defects configuration existing in intermetallic-intermetallic interface could generate back stress as well.

For nanolaminated intermetallics, strain transfer into the monolithic $Al_9(Fe,Co,Ni)_2$ could be very challenging, which necessitates a large back stress to drive defect activity. Under such a large back stress, SFs or other defects may be activated in intermetallic phases. Due to the limited plasticity of intermetallics, HDI stress stemming from intermetallic interfaces would increase rapidly during the initial loading process and remain saturated after plastic relaxation, which is consistent with experiment observations of back stress saturation for both regions (Fig. 7c).

Under the context of strength-ductility paradox for most metallic materials, some factors contribute to around 20% plasticity in these high-strength AM alloys as shown from both macropillar and micropillar compression tests. First, the Al matrix accommodates a majority

of plastic strain as verified by dislocations in Al in the deformed pillars. Second, the back stress from heterogeneous interfaces sustains significant work hardening. As discussed earlier, SFs and other defects have been observed in deformed nanoscale intermetallics to accommodate plasticity under high stresses. Third, the interfaces between the two nanoscale intermetallic phases may have increased the fracture strength in the fine rosette region, so that plastic yielding can occur before fracture. The improved fracture toughness of intermetallic nanolaminates is witnessed by microcracks restrained within lamellae in coarse rosettes, as shown in Fig. 8. This crack inhibition effect will release local stress concentration and delay catastrophic fracture.

In summary, a custom-made $Al_{92}Ti_2Fe_2Co_2Ni_2$ alloy was fabricated by LPBF. This alloy has rosettes of nanoscale intermetallics and a macroscopic engineering compressive strength exceeding 800 MPa and 20% plasticity. Micropillar compression tests reveal that the fine rosette regions can achieve a microscopic compressive strength of nearly 1.0 GPa and at least 15% plasticity. The simultaneous achievement of high strength and plasticity arises from the large back stress accommodated through heterogenous intermetallic nanolaminate interfaces. Significant plasticity was also observed in the medium entropy monoclinic $Al_9(Fe,Co,Ni)_2$ intermetallic phases. The mechanisms that trigger the formation of abundant stacking faults in monolithic $Al_9(Fe,Co,Ni)_2$ remain to be illuminated by future modeling investigations. Our results shed light on incorporation of nanoscale intermetallics rosettes in the design of ultra-strong Al alloys with prominent plasticity.

## Methods

### Powder processing and manufacturing

Spherical powder with a nominal composition of $Al_{92}Ti_2Fe_2Co_2Ni_2$ (at.%) satisfying $-53 + 15\,\mu m$ were gas atomized by Atlantic Equipment Engineering, Inc. Additive manufacturing was performed by using a laser powder bed fusion (LPBF) instrument, SLM 125 HL metal 3D printer in Argon atmosphere with the oxygen level below 1000 PPM. Printing was conducted by utilizing a 400 W IPG fiber laser ($\lambda = 1070\,nm$) with a laser power of 200–300 W, a scan speed of 1200 mm/s, a hatch space of 100 $\mu m$, a layer thickness of 30 $\mu m$ and a laser spot of 70 $\mu m$ in diameter. Build plate was preheated to 200 °C and each layer rotated by 67°. Cylindrical samples with height 12 mm and diameter 6 mm were fabricated for bulk compression tests. Cubic samples with dimensions $10 \times 10 \times 5$ mm were printed for microstructure characterization, nanoindentation and micropillar compression tests.

### Structural characterization

The microstructure of Al alloy was investigated by X-ray diffraction (XRD), scanning electron microscopy (SEM), transmission electron microscopy (TEM) and atom probe tomography (APT). Samples were mechanically grinded and polished down to 1 $\mu m$ diamond paste. XRD was performed on a PANalytical Empyrean X'pert PRO MRD diffractometer with a $2 \times Ge$ (220) hybrid monochromator to select Cu $K_{\alpha 1}$ in the $2\theta$-$\omega$ geometrical configuration. Scanning electron microscopy (SEM) experiments were performed by using a Thermo Fischer Quanta™ 3D and Teneo™ high-resolution Field Emission SEM microscopes with a back scattering detector operated at 30 kV. A Thermo Fisher Talos 200X TEM microscope with an acceleration voltage of 200 kV was utilized to capture bright field (BF), dark field (DF), scanning transmission electron microscopy (STEM) images, and Energy dispersive spectrometry (EDS) maps. Crystal orientation mapping was performed by using a NanoMEGAS detector. APT Samples were prepared using standard focused ion beam (FIB) lift-out procedures on a Scios 2 DualBeam FIB/SEM, followed by a series of annular milling steps with decreasing radii to achieve a tip radius of approximately 50 nm. Atom probe data were collected on a CAMECA LEAP 5000XS

APT, using both voltage and laser mode acquisition. For the former, a pulse fraction of 20%, temperature of 50 K, and a pulse rate of 200 kHZ were employed. For the latter, similar values for temperature and pulse rate were employed, with a laser pulse energy of 80 pJ to ensure complete field ion evaporation. Data reconstruction and analyses were conducted using AP Suite 6.1 software.

### Mechanical testing

Nanoindentation experiments were performed with a Bruker's Hysitron TI Premier nanoindenter with a Berkovich tip under displacement-control mode at 800 nm depth on well-polished samples. Hardness information was assessed from an area of $100 \times 100\,\mu m^2$ covering representative microscale features with 121 indents with 10 $\mu m$ spacing in both dimensions. Progressive indentation with multiple continuous loading-unloading segments at incremental penetration depths were conducted for each indentation. Hardness and Young's modulus were determined from an average of 10 measurements. Bulk compression tests were performed on an MTS framework with a 30 kN load cell and a strain rate of $10^{-3}\,s^{-1}$ after polishing and leveling the top and bottom surfaces of as-printed cylindrical samples for better alignment. In situ micropillar compression tests were performed in the Quanta™ 3D SEM microscope equipped with a Hysitron PI 88× R PicoIndenter and a real-time video recorder. Micropillars were produced by FIB, with the height of 10 $\mu m$,, the diameter of 5 $\mu m$, and an aspect ratio of 2:1. Both 10 and 20 $\mu m$ diamond flat-punch tips were used and strain rate was set as $5 \times 10^{-3}\,s^{-1}$. An average drift rate of 0.2–0.6 nm/s was determined for displacement correction.

## Data availability

The data supporting the findings of this study are available within the article and its supplementary Information. Additional data are available from the corresponding author on requests.

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

## Acknowledgements

This work is supported primarily by NSF-DMR-MMN 2210152 (X.Z.). Access to the Electron Microscopy Facility center at Purdue University is also acknowledged. The ASTAR crystal orientation system in TEM microscope is supported by ONR-DURIP award N00014-17-1-2921 (H.W.). HW acknowledge support by the U.S. Office of Naval Research (Contract No N00014-22-1-2160 for TEM). Atom probe tomography was performed at the Materials Research Laboratory at the University of Illinois at Urbana-Champaign using a CAMECA LEAP 5000-XS instrument purchased with support from the NSF under Grant No. DMR-1828450 (P.B.). K.C. and P.B. thank Dr. Amit Verma for his expert assistance with the collection and analysis of atom probe data.

## Author contributions

A.S. and B.S. conceived the idea and designed the experiments. B.S., J.L. and A.S. prepared the material and performed bulk compression tests. T.N., X.S. and A.S. executed nanoindentation. T.N., C.S. and A.S. conducted micropillar compression. P.B. and K.C. performed APT analyses and wrote the corresponding text. L.H., B.P.Z. and A.S prepared SEM samples. T.N., C.S. and A.S. fabricated FIB samples. T.N., C.S., Z.S., X.S. and A.S. contributes to SEM, TEM and STEM analyses. H.W. and X.Z. conceptualized and supervised the project. A.S. prepared the manuscript. B.S., J.L., T.N., C.S., Z.S., X.S., K.C., P.B. and X.Z. revised it. All the authors contributed to the discussion parts.

## Competing interests

X.Z., A.S., B.S. and H.W. were listed as inventors on a patent application (NO. 63/612,129) related to this work with X.Z. being the patent applicant. The composition was patented. All authors declare that they have no other competing interests.
