## [Peer Review File · Nature Communications]

Additive manufacturing of an ultrastrong, deformable Al alloy with nanoscale intermetallicsREVIEWER COMMENTS

Reviewer #1 (Remarks to the Author):

General Comments:

The authors present the characterization of an Al-TiFeCoNi alloy, additively manufactured with laser powder bed fusion. A detailed microstructural analysis revealed a rosette/lamella structure that has been owed to the high compressive strength recorded during bulk and micropillar testing. Overall, the work has merit and should be further investigated. However, the study does not feel complete to the point that the work is considered novel enough to be published in Nature Communications.

Microstructural investigations are important for fundamental understanding, but the current manuscript does not connect well with the processing or the performance side of the LPBF-processed Al-TiFeCoNi alloy. Not sure why the authors did not attempt tensile, fatigue, or creep type mechanical testing, since these are the best measures of mechanical performance. I would imagine that cracks or other detrimental defects may have formed during the printing process, which would have inhibited the author from performing the aforementioned tests, relying on only compression.

Additional Comments to Address:

1. I'm sure the authors are ready for this question, but why compression testing? Theoretically tensile and compression are mirrors of each other, but we know that's not necessarily true in reality.
2. Were there cracks or other major defects that formed during the printing process? The authors indicate performing their microstructural analysis on the sample printed with 300 W, but mechanical testing was performed on the sample printed with 200 W. Any particular reason? What was the major difference between the 200 and 300 W samples? Also, the parameter combination used is similar to AlSi10Mg, does that mean this alloy is highly printable?
3. Line 43: What do the authors mean by "...design and integration..."
4. Lines 81-82: Details on the melt pool size appear unnecessary. It's hard to judge the size looking at just the bulk macrostructure. You need to look at the last layer melted, which is unaffected by any additional layer fused on top.
5. Lines 122-124: I appreciate the relationships between phases. It appears that they are pretty coherent. Any measurement on coherency? May be important for understanding of nucleation and growth. How do the rosettes affect the grain structure (size, morphology, etc.)?
6. Lines 128-129: It appears that the Ni and Ti are near each other since Al₃X is favorable for those two elements. But, the chemical increase from 0.1 to 0.5 (measured by EDS) does not appear to really be significant. Is it significant? What exactly are the line scans in Figure 2 trying to show that can be better described with additional detail?
7. Lines 144-151: Not sure hardness is important here. There really doesn't appear to be much correlation in the hardness maps. I think the idea that the melt pool boundaries are "harder" is a little handwavy. The 3H approximation in Line 246 really doesn't add to the paper.
8. Line 153: You tested the 200 W sample, what about the 300 W sample?
9. Line 165: What particles are showing? It's hard to see exactly, but can you explain what the particles actually are?
10. Lines 216-217: So, lamella like the ones observed in the sample are common, especially when the common transition metals are added like Fe, Ni, Co, Cr. Not sure what the alternating structure is directly a result of the thermal cycling. Otherwise, the residual of the melt pool wouldn't be observable, since that is the as-solidified structure. Also, more mechanistic understanding of precipitate strengthening, I might take another look at the Al-Ce papers. (Henderson et al., 2021; Hyer et al., 2022; Martin et al., 2021; Plotkowski et al., 2017; Sisco et al., 2021)
11. Lines 257-258: Hard to say what heating the build plate to 200°C would do to the build. Since the solidification microstructure is present, I doubt any recovery has occurred. I think investigating and understanding the relationship between solidification and what was observed could be elucidated better in the discussion.
12. Line 272: Do you have a good reference on the formation of GNDs due to the mismatch in thermal expansion?

13. Line 296: Word choice needs to be reconsidered "Scattered prior study..."

Henderson, H. B., Hammons, J. A., Baker, A. A., McCall, S. K., Li, T. T., Perron, A., Sims, Z. C., Ott, R. T., Meng, F., Thompson, M. J., Weiss, D., & Rios, O. (2021). Enhanced thermal coarsening resistance in a nanostructured aluminum-cerium alloy produced by additive manufacturing. *Materials & Design*, 209, 109988. <https://doi.org/https://doi.org/10.1016/j.matdes.2021.109988>

Hyer, H., Mehta, A., Graydon, K., Kljestan, N., Knezevic, M., Weiss, D., McWilliams, B., Cho, K. Y., & Sohn, Y. (2022). High strength aluminum-cerium alloy processed by laser powder bed fusion. *Additive Manufacturing*, 52, 102657. <https://doi.org/ARTN 102657>
10.1016/j.addma.2022.102657

Martin, A. A., Hammons, J. A., Henderson, H. B., Calta, N. P., Nielsen, M. H., Cook, C. C., Ye, J., Maich, A. A., Teslich, N. E., & Li, T. T. (2021). Enhanced mechanical performance via laser induced nanostructure formation in an additively manufactured lightweight aluminum alloy. *Applied Materials Today*, 22, 100972.

Plotkowski, A., Rios, O., Sridharan, N., Sims, Z., Unocic, K., Ott, R. T., Dehoff, R. R., & Babu, S. S. (2017). Evaluation of an Al-Ce alloy for laser additive manufacturing. *Acta Materialia*, 126, 507-519. <https://doi.org/10.1016/j.actamat.2016.12.065>

Sisco, K., Plotkowski, A., Yang, Y., Leonard, D., Stump, B., Nandwana, P., Dehoff, R. R., & Babu, S. S. (2021). Microstructure and properties of additively manufactured Al-Ce-Mg alloys. *Scientific Reports*, 11(1), 6953. <https://doi.org/10.1038/s41598-021-86370-4>

Reviewer #2 (Remarks to the Author):

The authors have presented extensive characterization and discussion of mechanical properties of a novel Al-alloy for AM applications, which appears to make significant progress on the primary strength and ductility challenges facing this class of alloys in AM. I believe only minor issues should be addressed and generally consider the manuscript to be in good condition for publication. Detailed thoughts and comments follow below:

Figure 1: please note in the caption that Z is the build direction (is in the text already, but would be helpful here). May be helpful to put the axes in each subfigure

Figure 3: Can you explain ASTAR orientation mapping, I not familiar with this. It does look like you would have had to use TKD geometry for this kind of EBSD resolution but I don't see any mention of it in the results section.

Please check that you define all of your acronyms at their first use. I see you define many of these in the methods section but this is typically at the end of the article. Of course acronyms like SEM, TEM, STEM, EDS, BF, HAADF, etc. are common parlance for electron microscopists, but it is generally good practice to make sure you define all your acronyms, as you have done already for some terms in the main text.

Figure 3. Scale bar on the EBSD in particular is quite hard to see.

Figure 4. Scale bars here are basically impossible to see on the the TEM image (subfigure a) and the interphase plane. Might be good to make it clear in the caption that the interphase plane is from a different tip. Do you have a similar TEM image of it as subfigure a) prior to evaporation?

Rosettes are typically understood to form due to liquid entrapment and high undercooling necessitating secondary nucleation events for solidification, with length scales correlated to the cooling rates experienced during solidification. Did you look into this at all with some simple thermal modelling? I am not sure of the temperature-dependence of surface tension for this family of alloys

but Marangoni flows seem less likely to be a strong influencing factor here as you so clearly have fine rosettes near melt pool boundaries (highest thermal gradients) and coarser rosettes closer to the center (highest interface velocities).

S6 seems like an important figure and you should definitely provide citations for the data of other alloys, as it does not appear that these were collected as part of this work. These were all collected for AM alloys in compression, correct?

Do you have a limit on figures? It seems like some of the supplementary figures are sufficiently important to be included in the main manuscript, in particular S2 and S6.

May be worth trying to approximate GND content from your EBSD data using the Nye tensor approximation to estimate its contribution to strengthening.

Nice microscopy all around. Would be very interested to see the performance of these alloys in tension as well in future work. On a related note, it may be helpful to talk about bulk defect structures observed in these alloys. Did you observe any macroscopic cracking/hot tearing in these alloys? What was the bulk porosity of the parts you printed?

Reviewer #1 (Remarks to the Author):

General Comments:

The authors present the characterization of an Al-TiFeCoNi alloy, additively manufactured with laser powder bed fusion. A detailed microstructural analysis revealed a rosette/lamella structure that has been owed to the high compressive strength recorded during bulk and micropillar testing. Overall, the work has merit and should be further investigated. However, the study does not feel complete to the point that the work is considered novel enough to be published in Nature Communications. Microstructural investigations are important for fundamental understanding, but the current manuscript does not connect well with the processing or the performance side of the LPBF-processed Al-TiFeCoNi alloy. Not sure why the authors did not attempt tensile, fatigue, or creep type mechanical testing, since these are the best measures of mechanical performance. I would imagine that cracks or other detrimental defects may have formed during the printing process, which would have inhibited the author from performing the aforementioned tests, relying on only compression.

Answers: We greatly appreciate the reviewer's questions and comments. Many of these comments addressed the detailed questions raised by the reviewer in the following section.

The current manuscript highlighted a high flow strength, good deformability under compression and the underlying novel deformation mechanisms in AM Al alloys. We agree that further mechanical testing, such as tension and creep, would be very interesting, and we are investigating the AM alloys along these directions.

Also, we would like to emphasize the originality and scientific contributions of this work. First,

this Al alloy exhibits ultrahigh mechanical strength and compressive plasticity, making it probably one of the strongest Al alloys manufactured by 3D printing to date. Second, a prominent back stress discovered from micropillar compression tests, indicating back stress would play an important role in strengthening and toughening in the AM Al alloys. Third, plastic deformability rendered by dislocations and stacking faults is discovered, for the first time, in the brittle monoclinic Al_9Co_2 type intermetallics, which would explain the compressive plastic deformability of the AM Al alloys at high stress.

Additional Comments to Address:

1. I'm sure the authors are ready for this question, but why compression testing? Theoretically tensile and compression are mirrors of each other, but we know that's not necessarily true in reality.

Answer: This is a great question. This AM Al alloy exhibits a combination of high compressive strength and large plastic deformability. To address the reviewers' comments, we have performed tension tests. Fig. R1 shows the stress-strain curves of the 200 W dogbone samples tested via the in-situ SEM tensile stage both at room and elevated temperatures. The high-strength AM Al alloy has limited tensile ductility but a high strength even up to 500 °C. The limited tensile ductility is likely due to fracture of intermetallics in the rosettes, rather than the fabrication defects (cracks or porosity).

Fig. R1 Tensile stress - strain curves for samples printed with 200 W tested at both room and elevated temperatures.

The asymmetry in tension-compression behaviors may arise from different deformation mechanisms. Without the confinement effect imposed by compressive stress, co-deformation mediated by dislocations and stacking faults in the intermetallics is hard to activate. Instead, the delamination of intermetallic-Al interface would lead to pre-mature fracture in tension.

2.1 Were there cracks or other major defects that formed during the printing process?

Answer: We appreciate the physical insights from the reviewer. We found no cracking or cavity in the as-printed state (see Fig. R2).

Hot cracking is found to be one of the most concerned defects for SLM Aluminum alloys that jeopardizing the mechanical performances. However, the authors managed to tackle the printability issue by using a compliant, sacrificial, scaffold support structure, and achieved

crack-free fully dense specimens in the as-printed state, examined by cross-section optical images and X-ray micro-CT (XCT). This long exploration is packaged into a separate story focusing on crack elimination and the manuscript is under review at another journal. Two representative samples printed with the same parameter sets (scanning speed = 1200mm/s, laser power = 250W) are analyzed by XCT as shown in Fig. R2. Cracks grow vertically upward in the control sample (Fig. R2a-c), whereas a crack-free region is revealed above the interface for the support sample (Fig. R2d-f). Cracks are confined in the porous support (template) region without extending into the actual (solid) part.

Fig. R2. X-ray micro-CT (XCT) analyses comparing samples printed (a-c) without support and (d-f) with the support. (a) Cross-sectional XCT image showing cracks in sample without support. (b) Back-forward and (c) bottom-up projection views showing abundant continuous cracks. (d) Cross-sectional XCT image and (e) Back-forward projection view showing crack free region in samples with support. (f) Bottom-up view showing cracks initiated from support/built plate interfaces. Inserts on the upper right corner of each figure demonstrate the field of view with a red rectangle in relation to the whole specimen geometry.

As shown in Fig. R3, compared with the control sample with a porosity of 0.19%, the support

sample has a higher relative density (porosity less than 0.004%), and a smaller average pore diameter. These two figures demonstrate the success in eliminating cracks and porosity in the as-printed Al alloys.

Fig. R3 X-ray micro-CT analyses on the porosity measurements for specimens printed without the support (a) and with the support (b). (c) Lognormal-type pore size distributions for these two samples and the denoted average pore diameters. C – control (no support); S – with support.

2.2. The authors indicate performing their microstructural analysis on the sample printed with 300 W, but mechanical testing was performed on the sample printed with 200 W. Any particular reason?

Answer: Fig. R4 shows that the macroscopic compression tests on specimens printed with 200, 250 and 300W have comparable strength and plasticity. Hence, micropillar compression tests focused on the specimens printed at 300 W. More specifically, our micropillar compression tests (Fig. 6 in the manuscript, Fig. R4 shown below) show that the flow stress of specimens decreases with increasing laser power. A similar trend has been reported in AM 718 Ni alloys, where the strength difference arises from grain sizes¹.

Fig. R4 (Fig. 6 in the manuscript) (a) Engineering stress – engineering strain curves for bulk compression tests performed on the as-printed $\text{Al}_{92}\text{Ti}_2\text{Fe}_2\text{Co}_2\text{Ni}_2$ pillars with schematic diagrams showing the geometry of specimen and the typical barreling phenomenon after compression for one of the deformed specimens (200W). Numbers in the legend denote laser power. (b) True stress - true strain curves superimposed with working hardening rates showing 7% uniform compression in the early stages of deformation.

2.3. What was the major difference between the 200 and 300 W samples?

Answer: We analyzed the microstructure on all the three samples (printed at 200, 250 and 300W) and found them comparable. As shown from XRD profiles (Fig. R5), phase constituents and fractions do not change significantly with volumetric energy density (VED). All these three conditions give characteristic fine rosette and coarse rosette regions.

Fig. R5 X-Ray Diffraction (XRD) profiles of powder and selected samples with different VEDs. No substantial difference in phases is observed.

2.4. Also, the parameter combination used is similar to AlSi10Mg, does that mean this alloy is highly printable?

Answer: The parameter sets used here were optimized from a larger parameter matrix. This alloy is printable over a relatively wide range of laser power, allowing us to achieve tunable mechanical performances.

3. Line 43: What do the authors mean by “...design and integration...”

Answer: Thank you for the catch. The original sentence is rephrased as follows.

“To fulfill the complex geometrical constraints for industrial applications, selective laser melting (SLM) has been increasingly used to fabricate parts of Al alloys, offering a high level

of design flexibility.”

4. Lines 81-82: Details on the melt pool size appear unnecessary. It’s hard to judge the size looking at just the bulk macrostructure. You need to look at the last layer melted, which is unaffected by any additional layer fused on top.

Answer: The authors appreciate the suggestion and comments. A more systematic analysis on the melt pool dimensions was done by single laser track experiments. Those data are categorized into the printability paper submitted elsewhere. Here the dimensions are more for documentation purposes.

5. Lines 122-124: I appreciate the relationships between phases. It appears that they are pretty coherent. Any measurement on coherency? May be important for understanding of nucleation and growth. How do the rosettes affect the grain structure (size, morphology, etc.)?

Answer: The authors highly appreciate the reviewer’s insights. The coherency (lattice matching) was observed from selected area diffraction patterns.

The followings have been added to the revised manuscript.

“The lattice mismatch estimated from diffraction patterns for these matching planes are 2.5% and 2.1%, respectively.”

There is indeed some lattice matching between Al_3Ti and $\text{Al}_9(\text{Fe,Co,Ni})_2$. Such lattice match may reduce the interfacial energy and promote the formation of nanolaminated intermetallics. If the quasi-equilibrium condition holds true during nucleation and growth, the crystal will exhibit anisotropic Wulff shapes with minimal total interfacial energy. The coherency between Al_3Ti and $\text{Al}_9(\text{Fe,Co,Ni})_2$ does refine the lamellar thickness for the intermetallic cores. The

interfacial energy is proportional to mismatch strain. Therefore, a smaller excessive Gibbs free energy by a small mismatch strain allows the formation of refined intermetallic rosettes with larger interfacial area.

The formation of high melting temperature intermetallics in the melt pools may promote heterogeneous nucleation of Al during subsequent solidification and thus reduce the grain size of Al. Similar effects have been reported for Al alloys with the aid of nucleants, such as $\text{Al}_3\text{Zr}^{2-4}$, $\text{Al}_3\text{Sc}^{5,6}$, Al_3Ti^7 , TiB_2^{8-11} , for the refinement of grain size of Al matrix. We found that the fine rosettes enable the formation of smaller grain sizes in the melt pools, while the coarse rosettes tends to be surrounded by large Al grains.

The followings have been added to the revised manuscript.

“The lattice matching between Al_3Ti and $\text{Al}_9(\text{Fe,Co,Ni})_2$ may reduce the interfacial energy, and promote the formation of nanolaminated intermetallics. Under the assumption of quasi-equilibrium condition, the crystal will exhibit anisotropic Wulff shapes to reduce interfacial energy, promoting the formation of refined intermetallic rosettes with larger interfacial area. The formation of high melting temperature intermetallics in the melt pools may promote heterogeneous nucleation of Al grains during subsequent solidification and thus reduce the grain size of Al. Similar effects have been reported for Al alloys with the aid of nucleants, such as $\text{Al}_3\text{Zr}^{2-4}$, $\text{Al}_3\text{Sc}^{5,6}$, Al_3Ti^7 , TiB_2^{8-11} , for the refinement of grain size of Al matrix.”

6. Lines 128-129: It appears that the Ni and Ti are near each other since Al_3X is favorable for those two elements. But, the chemical increase from 0.1 to 0.5 (measured by EDS) does not appear to really be significant. Is it significant? What exactly are the line scans in Figure 2 trying to show that can be better described with additional detail?

Answer: The authors appreciate the review's observations and comments. In general, the transition metal elements have very low solid solubility in Al: Ti < 0.05 at%, Fe: < 0.04 at%, Co < 0.05 at%, Ni < 0.04 at%. Hence the APT composition analysis in Fig. 4c suggests that Ni and Ti concentration in Al (0.1-0.6 at.%), have largely exceeded their equilibrium solid solubility, as well as the weak segregation of TM elements at grain boundaries. Supersaturation may arise from the rapid solidification process. More importantly, APT data (Fig. 4d-f) obtained near the Al/Al₉(Fe,Co,Ni)₂ interface, taken from a different tip, confirm the nearly equal content in Fe, Co and Ni, and the absence of Ti. Lastly the TM elements do not display any significant clustering, as illustrated in Fig. 4(f), indicating that these elements are well mixed.

The TEM EDS maps and line scans in Fig. 2 are used to identify the phase distribution and stoichiometry. The fine rosettes are composed of Al₃Ti cores surrounded by laminated Al₃Ti and Al₉(Fe,Co,Ni)₂. The theoretical Al content ranges from 0.75 (for Al₃Ti) to 0.82 (for Al₉Co₂), which is comparable to the measured value from 0.74 to 0.86 in Fig. 2c. The cellular wall in the coarse rosette region is composed of Al₉(Fe,Co,Ni)₂ along with Al grains.

The manuscript has been revised as follows:

“In general, the transition metal elements have low solid solubility in Al: Ti < 0.05 at%, Fe: < 0.04 at%, Co < 0.05 at%, Ni < 0.04 at%. Hence the APT composition analysis in Fig. 4c suggests that Ni and Ti concentration in Al (0.1-0.6 at.%), has largely exceeded their equilibrium solid solubility, presumably due to supersaturation from rapid solidification.”

7. Lines 144-151: Not sure hardness is important here. There really doesn't appear to be much correlation in the hardness maps. I think the idea that the melt pool boundaries are “harder” is a little handwavy. The 3H approximation in Line 246 really doesn't add to the paper.

Answer: The authors thank the reviewer for comments. Our indentation studies show that the hardness map correlates with the melt pool boundaries. The melt pool boundaries are populated with a higher fraction of intermetallics (97%) which are mostly fine rosettes, and thus these regions are harder than the melt pool interior with a lower fraction of intermetallics in form of coarse rosettes (40%). The indentation hardness and flow stress measured from micropillar compression tests follows the classical 3 times relationship. Such statement is removed as suggested by the reviewer.

8. Line 153: You tested the 200 W sample, what about the 300 W sample?

Answer: The 200 W, 250 W and 300 W samples were all examined by macro-pillar compression tests (Please see line 157 - 158 and Fig. 6a). Our studies show that a lower laser power led to a higher flow strength and a slightly greater plastic deformability.

9. Line 165: What particles are showing? It's hard to see exactly, but can you explain what the particles actually are?

Answer: The authors refer to the rosette precipitates as “particles” appearing on the deformed pillar surface. To avoid ambiguity, we changed the word “particles” into “rosette precipitates”. The rosette precipitates protruded from the pillar surface during deformation.

The manuscript has been revised as follows:

“Rosette precipitates were readily visible on the pillar surfaces in the coarse rosette region due to axial compression, while a wrinkled and wavy surface shows less apparent protrusion in the fine rosette region. For the coarse rosettes, the evident rosette precipitate extrusions to the outer surfaces manifest the preferential plastic flow in the soft aluminum matrix and insignificant

plasticity in the hard intermetallics precipitates.”

10. Lines 216-217: So, lamella like the ones observed in the sample are common, especially when the common transition metals are added like Fe, Ni, Co, Cr. Not sure what the alternating structure is directly a result of the thermal cycling. Otherwise, the residual of the melt pool wouldn't be observable, since that is the as-solidified structure. Also, more mechanistic understanding of precipitate strengthening, I might take another look at the Al-Ce papers. (Henderson et al., 2021; Hyer et al., 2022; Martin et al., 2021; Plotkowski et al., 2017; Sisco et al., 2021)

Answer:

(1) The authors appreciate the valuable comments from the reviewer. Sorry for the confusion on the usage of “thermal cycle”. Here is what we meant to discuss. First, we envision that the combinations of cooling rate gradient and solute gradient in the melt pool lead to the formation of alternating fine and coarse rosettes. A higher fraction of fine intermetallics resides near the bottom of melt pools, presumably due to a high cooling rate and sufficient supply of transition metal solutes. In comparison, the melt pool interior has primarily coarse rosettes due to a relatively reduced cooling rate and depleted transition metal solutes in the melt pool. To examine this hypothesis, we confirmed the existence of heterogeneous melt pool structure by performing the single laser track experiment (please see Fig. R5a, b). Interestingly, in some cases, the Marangoni flow may have induced prominent convection in the melt pool, leading to the distribution of striated fine rosettes (Fig. R5b) in AM Al alloys printed at 100 W. Similar phenomenon was also observed in Fig. 1c for Al alloys printed at 300 W.

Fig. R5 SEM images for representative cross sections of the AM Al alloys from selected single laser track experiments. Processing parameters are (a) $P = 370 \text{ W}$, $v = 3000 \text{ mm/s}$; (b) $P = 100 \text{ W}$, $v = 600 \text{ mm/s}$. Arrows in (b) indicate the possible convection flux directions driven by Marangoni flow.

(2) Thank you very much for providing additional references. We have added these references to the revised manuscript and incorporate them to the discussion on strengthening mechanisms.

The manuscript has been revised as follows.

“There are limited studies showing the formation of complex intermetallics in AM Al alloys¹²⁻¹⁶. Some prior studies also suggest that complex metallic compound could deform by introducing metadislocations^{17,18}, which rarely exist in high symmetry metallic materials, and metadislocations have been reported in $\text{Al}_{13}\text{Co}_4$ with monoclinic crystal structure¹⁹.”

11. Lines 257-258: Hard to say what heating the build plate to 200°C would do to the build. Since the solidification microstructure is present, I doubt any recovery has occurred. I think investigating and understanding the relationship between solidification and what was observed could be elucidated better in the discussion.

Answer: The authors appreciate the comments from the reviewer. The as-solidified structure is mainly composed of intermetallic-skeleton and cellular Al. The manuscript has been revised as follows:

“Dislocations in the as-printed state do not play a significant role in strengthening the alloy. TEM experiments (Fig. S7) show a moderate dislocation density $4.7 \times 10^{13} \text{ m}^{-2} \sim 1.0 \times 10^{14} \text{ m}^{-2}$. Therefore, strengthening contribution from these dislocations can be estimated to be 25 - 36 MPa by using:

$$\sigma_{dist} = \beta M G b \sqrt{\rho_{dist}},$$

where $\beta = 0.16$, $M = 3.06$, $G = 26.5 \text{ GPa}$ and $b = 0.286 \text{ nm}^{20,21}$.”

Fig. R6 Dislocation density estimation from TEM/STEM micrographs. (a) STEM image on several cellular Al grains in Region 1. (b) TEM image on one Al grain in Region 2 and (c) the corresponding STEM image on the central area showing better contrast for dislocations.

12. Line 272: Do you have a good reference on the formation of GNDs due to the mismatch in thermal expansion?

Answer: The authors highly appreciate the review’s question. Dislocation density estimated from TEM images in two regions are $1.0 \times 10^{14} \text{ m}^{-2}$ and $4.7 \times 10^{13} \text{ m}^{-2}$. These dislocations include both statistically stored dislocations (SSDs) and geometrically necessary dislocations (GNDs). GNDs present in the as-printed state are induced from either CTE mismatch or accommodation on the heterogeneous deformation under cyclic thermal stress. So GNDs resulting from CTE mismatch would be a portion of the total dislocations. Dislocations or

GNDs in the as-printed state do not play a significant role in strengthening the alloy.

The following references regarding GND formation due to the mismatch in CTE have been added to the manuscript.

[1] S. Liu, X. Wang, Q. Zu, B. Han, X. Han, C. Cui, Significantly improved particle strengthening of Al–Sc alloy by high Sc composition design and rapid solidification, *Materials Science and Engineering: A*. 800 (2021) 140304. <https://doi.org/10.1016/j.msea.2020.140304>.

[2] C.S. Goh, J. Wei, L.C. Lee, M. Gupta, Properties and deformation behaviour of Mg–Y2O3 nanocomposites, *Acta Materialia*. 55 (2007) 5115–5121. <https://doi.org/10.1016/j.actamat.2007.05.032>.

13. Line 296: Word choice needs to be reconsidered “Scattered prior study...”

Henderson, H. B., Hammons, J. A., Baker, A. A., McCall, S. K., Li, T. T., Perron, A., Sims, Z. C., Ott, R. T., Meng, F., Thompson, M. J., Weiss, D., & Rios, O. (2021). Enhanced thermal coarsening resistance in a nanostructured aluminum-cerium alloy produced by additive manufacturing. *Materials & Design*, 209, 109988.

<https://doi.org/https://doi.org/10.1016/j.matdes.2021.109988>

Hyer, H., Mehta, A., Graydon, K., Kljestan, N., Knezevic, M., Weiss, D., McWilliams, B., Cho, K. Y., & Sohn, Y. (2022). High strength aluminum-cerium alloy processed by laser powder bed fusion. *Additive Manufacturing*, 52, 102657. <https://doi.org/ARTN 102657>

[10.1016/j.addma.2022.102657](https://doi.org/10.1016/j.addma.2022.102657)

Martin, A. A., Hammons, J. A., Henderson, H. B., Calta, N. P., Nielsen, M. H., Cook, C. C., Ye, J., Maich, A. A., Teslich, N. E., & Li, T. T. (2021). Enhanced mechanical performance via laser induced nanostructure formation in an additively manufactured lightweight aluminum alloy. *Applied Materials Today*, 22, 100972.

Plotkowski, A., Rios, O., Sridharan, N., Sims, Z., Unocic, K., Ott, R. T., Dehoff, R. R., & Babu, S. S. (2017). Evaluation of an Al-Ce alloy for laser additive manufacturing. *Acta*

Materialia, 126, 507-519. <https://doi.org/10.1016/j.actamat.2016.12.065>

Sisco, K., Plotkowski, A., Yang, Y., Leonard, D., Stump, B., Nandwana, P., Dehoff, R. R., & Babu, S. S. (2021). Microstructure and properties of additively manufactured Al–Ce–Mg alloys. *Scientific Reports*, 11(1), 6953. <https://doi.org/10.1038/s41598-021-86370-4>

Answer: Thank you for the suggestion. We have added these references to the revised manuscript. The manuscript has been revised as follows.

“There are limited studies showing the formation of complex intermetallics in AM Al alloys^{12–16}. Some prior studies also suggest that complex metallic compound could deform by introducing metadislocations^{17,18}, which rarely exist in high symmetry metallic materials, and metadislocations have been reported in Al₁₃Co₄ with monoclinic crystal structure¹⁹.”

Reviewer #2 (Remarks to the Author):

The authors have presented extensive characterization and discussion of mechanical properties of a novel Al-alloy for AM applications, which appears to make significant progress on the primary strength and ductility challenges facing this class of alloys in AM. I believe only minor issues should be addressed and generally consider the manuscript to be in good condition for publication. Detailed thoughts and comments follow below:

1. Figure 1: please note in the caption that Z is the build direction (is in the text already, but would be helpful here). May be helpful to put the axes in each subfigure

Answer: Thank you for the suggestion. We have made corrections accordingly.

2. Figure 3: Can you explain ASTAR orientation mapping, I not familiar with this. It does look like you would have had to use TKD geometry for this kind of EBSD resolution but I don't see any mention of it in the results section.

Answer: The authors appreciate the reviewer's comments. ASTAR is a TEM-based automated crystal orientation mapping technique by using precession electron diffraction technique. By analyzing precise electron diffraction per pixel, it can generate inverse pole figures with comparable resolution to TKD results. Please see the following references as examples^{22,23}.

[1] Su, R., Neffati, D., Cho, J., Shang, Z., Zhang, Y., Ding, J., ... & Zhang, X. (2021). High-strength nanocrystalline intermetallics with room temperature deformability enabled by nanometer thick grain boundaries. *Science Advances*, 7(27), eabc8288.

[2] Sun, T., Shang, Z., Cho, J., Ding, J., Niu, T., Zhang, Y., ... & Zhang, X. (2021). Ultra-fine-grained and gradient FeCrAl alloys with outstanding work hardening capability. *Acta Materialia*, 215, 117049.

3. Please check that you define all of your acronyms at their first use. I see you define many of these in the methods section but this is typically at the end of the article. Of course acronyms like SEM, TEM, STEM, EDS, BF, HAADF, etc. are common parlance for electron microscopists, but it is generally good practice to make sure you define all your acronyms, as you have done already for some terms in the main text.

Answer: We appreciate the reviewer's comment on improving the readability. We have defined the acronyms accordingly from the beginning of the revised manuscript.

4. Figure 3. Scale bar on the EBSD in particular is quite hard to see.

Answer: Sorry for the inconvenience. We have enlarged the scale bar to improve readability.

5. Figure 4. Scale bars here are basically impossible to see on the TEM image (subfigure a) and the interphase plane. Might be good to make it clear in the caption that the interphase plane is from a different tip. Do you have a similar TEM image of it as subfigure a) prior to evaporation?

Answer: We have made changes accordingly. The 2nd row of APT results are indeed obtained from a different tip. We have added clarification in the revised manuscript. We regret not to be able to provide a similar TEM image for the APT tip of the interphase plane. For this second tip, a standard atom probe coupon with Si micro-posts [CAMECA Instruments Inc.] was utilized to prepare the tip, and with this method, APT tips cannot be transferred to a TEM holder. However, it was assessed that TEM imaging was not critical for this tip. Information regarding crystallographic poles was collected during APT data acquisition and this information has been used to improve the atomic reconstruction, specifically by providing a direct quantification of the image compression factor²⁴. In contrast, for the first tip (Figure 4a,b,c), a stainless-steel

grid with a specialized holder [Microscopy Supplies Australia] was used; this holder can be mounted onto a JEOL microscope single tilt holder, thus allowing the capturing of TEM images of the tip prior to atom probe acquisition, important to identify grain boundaries.

6. Rosettes are typically understood to form due to liquid entrapment and high undercooling necessitating secondary nucleation events for solidification, with length scales correlated to the cooling rates experienced during solidification. Did you look into this at all with some simple thermal modelling? I am not sure of the temperature-dependence of surface tension for this family of alloys but Marangoni flows seem less likely to be a strong influencing factor here as you so clearly have fine rosettes near melt pool boundaries (highest thermal gradients) and coarser rosettes closer to the center (highest interface velocities).

Answer: We highly appreciate the reviewer's insights on the solidification history of these complex intermetallic rosettes. The authors agree that it is the combination of cooling rate gradient and transition metal solutes gradient that jointly impacts the nucleation and growth of laminated fine/coarse rosettes. Although the fine rosettes are mostly distributed along melt pool boundaries, in some cases, we indeed observed that striated fine rosettes in the melt pool interior (please see Fig.1c, Fig. R5a,b and responses to question # 11 from the Reviewer 1). These findings suggest that these "misplaced" fine rosettes might be circulated by convection flows (Marangoni flow).

We have attempted to calculate the cooling rate from the lamellar spacing of the fine rosettes. However, the facts that many physical properties for two intermetallics are missing making the estimation very difficult and unreliable. Such an aspect will be examined in our future studies.

7. S6 seems like an important figure and you should definitely provide citations for the data of other alloys, as it does not appear that these were collected as part of this work. These were all collected for AM alloys in compression, correct?

Do you have a limit on figures? It seems like some of the supplementary figures are sufficiently important to be included in the main manuscript, in particular S2 and S6.

Answer: We appreciate the reviewer's comments. Nature Communications have a figure limit of 10 figures/charts/tables, so we could not include both S2 and S6 in the main body of the manuscript.

Yes, in Fig. S6, all the results were acquired under uniaxial compression for bulk samples. The citations for the supplementary information are listed at the end of supplemental information.

8. May be worth trying to approximate GND content from your EBSD data using the Nye tensor approximation to estimate its contribution to strengthening.

Answer: The authors highly appreciate the reviewer's insights on dislocation density estimation. As for the dislocation density in Al grains and its contribution to strengthening, one estimation based on TEM/STEM images is presented on Comment 11 from the Reviewer 1. As for the GNDs in intermetallics, here is a Kernel Average Misorientation (KAM) map showing the relative GND density (Fig. R7c). Al grains surrounding rosette exteriors in blue color indicate formation of few GNDs while rosette interiors have speckled yellow/red pixels, implying the existence of high-density GND. These qualitative results are aligned with electron microscope examinations. However, due to the similarity of Al_3Ti to Al and alike diffraction patterns of Al_9X_2 in all the orientations, GND density approximated from KAM map may be inaccurate.

Fig. R7 Inverse pole figures for (a) Al/Al₃Ti and (b) Al₉(Fe,Co,Ni)₂ type intermetallics. (c) KAM map showing the relative GND density.

9. Nice microscopy all around. Would be very interested to see the performance of these alloys in tension as well in future work. On a related note, it may be helpful to talk about bulk defect structures observed in these alloys. Did you observe and macroscopic cracking/hot tearing in these alloys? What was the bulk porosity of the parts you printed?

Answer: Thank you for the comments and suggestions. This alloy exhibits excellent thermal stability, high flow stress, but relatively limited ductility under tension. Please see Fig. R1 for tension data.

We have successfully solved the hot tearing/large cavities issues. Fig. R2 and Fig. R3 show the as-printed AM Al alloys are crack-free and almost fully dense, which alleviates the influence of defects on plastic deformability. Porosity measured from Mirco-CT is below 0.1%. Some tension data, defects and porosity are discussed in response to Question #2 from Reviewer 1.

References:

1. Stegman, B. *et al.* Volumetric energy density impact on mechanical properties of additively manufactured 718 Ni alloy. *Materials Science and Engineering: A* **854**, 143699 (2022).
2. Nie, X. *et al.* Effect of Zr content on formability, microstructure and mechanical properties of selective laser melted Zr modified Al-4.24Cu-1.97Mg-0.56Mn alloys. *Journal of Alloys and Compounds* **764**, 977–986 (2018).
3. Croteau, J. R. *et al.* Microstructure and mechanical properties of Al-Mg-Zr alloys processed by selective laser melting. *Acta Materialia* **153**, 35–44 (2018).
4. Martin, J. H. *et al.* 3D printing of high-strength aluminium alloys. *Nature* **549**, 365–369 (2017).
5. Schmidtke, K., Palm, F., Hawkins, A. & Emmelmann, C. Process and Mechanical Properties: Applicability of a Scandium modified Al-alloy for Laser Additive Manufacturing. *Physics Procedia* **12**, 369–374 (2011).
6. Spierings, A. B., Dawson, K., Voegtlin, M., Palm, F. & Uggowitzer, P. J. Microstructure and mechanical properties of as-processed scandium-modified aluminium using selective laser melting. *CIRP Annals* **65**, 213–216 (2016).
7. Tan, Q. *et al.* Inoculation treatment of an additively manufactured 2024 aluminium alloy with titanium nanoparticles. *Acta Materialia* **196**, 1–16 (2020).
8. Wang, P. *et al.* A heat treatable TiB₂/Al-3.5Cu-1.5Mg-1Si composite fabricated by selective laser melting: Microstructure, heat treatment and mechanical properties. *Composites Part B: Engineering* **147**, 162–168 (2018).
9. Liu, Y. *et al.* Microstructural evolution and mechanical performance of in-situ TiB₂/AlSi10Mg composite manufactured by selective laser melting. *Journal of Alloys and Compounds* **853**, 157287 (2021).
10. Wang, J. *et al.* Selective laser melting of high-strength TiB₂/AlMgScZr composites:

- microstructure, tensile deformation behavior, and mechanical properties. *Journal of Materials Research and Technology* **16**, 786–800 (2022).
11. Xi, L., Guo, S., Gu, D., Guo, M. & Lin, K. Microstructure development, tribological property and underlying mechanism of laser additive manufactured submicro-TiB₂ reinforced Al-based composites. *Journal of Alloys and Compounds* **819**, 152980 (2020).
 12. Hyer, H. *et al.* High strength aluminum-cerium alloy processed by laser powder bed fusion. *Additive Manufacturing* **52**, 102657 (2022).
 13. Plotkowski, A. *et al.* Evaluation of an Al-Ce alloy for laser additive manufacturing. *Acta Materialia* **126**, 507–519 (2017).
 14. Sisco, K. *et al.* Microstructure and properties of additively manufactured Al–Ce–Mg alloys. *Sci Rep* **11**, 6953 (2021).
 15. Henderson, H. B. *et al.* Enhanced thermal coarsening resistance in a nanostructured aluminum-cerium alloy produced by additive manufacturing. *Materials & Design* **209**, 109988 (2021).
 16. Martin, A. A. *et al.* Enhanced mechanical performance via laser induced nanostructure formation in an additively manufactured lightweight aluminum alloy. *Applied Materials Today* **22**, 100972 (2021).
 17. Feuerbacher, M. & Heggen, M. On the concept of metadislocations in complex metallic alloys. *Philosophical Magazine* **86**, 935–944 (2006).
 18. Heidelmann, M., Heggen, M., Dwyer, C. & Feuerbacher, M. Comprehensive model of metadislocation movement in Al₁₃Co₄. *Scripta Materialia* **98**, 24–27 (2015).
 19. Heggen, M. & Feuerbacher, M. Core Structure and Motion of Metadislocations in the Orthorhombic Structurally Complex Alloy Al₁₃Co₄. *Materials Research Letters* **2**, 146–151 (2014).
 20. Hadadzadeh, A., Baxter, C., Amirkhiz, B. S. & Mohammadi, M. Strengthening

- mechanisms in direct metal laser sintered AlSi10Mg: Comparison between virgin and recycled powders. *Additive Manufacturing* **23**, 108–120 (2018).
21. Hu, Z. *et al.* Aging responses of an Al-Cu alloy fabricated by selective laser melting. *Additive Manufacturing* 101635 (2020) doi:10.1016/j.addma.2020.101635.
 22. Su, R. *et al.* High-strength nanocrystalline intermetallics with room temperature deformability enabled by nanometer thick grain boundaries. *Science Advances* **7**, eabc8288 (2021).
 23. Sun, T. *et al.* Ultra-fine-grained and gradient FeCrAl alloys with outstanding work hardening capability. *Acta Materialia* **215**, 117049 (2021).
 24. Vurpillot, F., Gault, B., Geiser, B. P. & Larson, D. J. Reconstructing atom probe data: A review. *Ultramicroscopy* **132**, 19–30 (2013).

REVIEWERS' COMMENTS